# TABX: A High-Throughput Sandbox Battle Simulator
# for Multi-Agent Reinforcement Learning

Hayeong Lee [* 1]   JunHyeok Oh [* 1]   Byung-Jun Lee [1 2]

## Abstract

The design of environments plays a critical role in shaping the development and evaluation of cooperative multi-agent reinforcement learning (MARL) algorithms. While existing benchmarks highlight critical challenges, they often lack the modularity required to design custom evaluation scenarios. We introduce the Totally Accelerated Battle Simulator in JAX (TABX), a high-throughput sandbox designed for reconfigurable multi-agent tasks. TABX provides granular control over environmental parameters, permitting a systematic investigation into emergent agent behaviors and algorithmic trade-offs across a diverse spectrum of task complexities. Leveraging JAX for hardware-accelerated execution on GPUs, TABX enables massive parallelization and significantly reduces computational overhead. By providing a fast, extensible, and easily customized framework, TABX facilitates the study of MARL agents in complex structured domains and serves as a scalable foundation for future research. Our code is available at: https://github.com/ku-dmlab/TABX.

## 1. Introduction

Deep reinforcement learning (RL) has become a powerful framework for solving complex sequential decision-making problems in environments involving multiple interacting agents. In such multi-agent settings, agents learn to cooperate, compete, or both, giving rise to challenges that do not arise in single-agent RL. To facilitate research in this direction, a variety of multi-agent environments have been developed (Samvelyan et al., 2019; Carroll et al., 2019; Kurach

*Equal contribution [1]Department of Artificial Intelligence, Korea University, Seoul, Republic of Korea [2]Gauss Labs Inc., Seoul, Republic of Korea. Correspondence to: Byung-Jun Lee <byungjun-lee@korea.ac.kr>.

*Proceedings of the 43$^{rd}$ International Conference on Machine Learning*, Seoul, South Korea. PMLR 306, 2026. Copyright 2026 by the author(s).

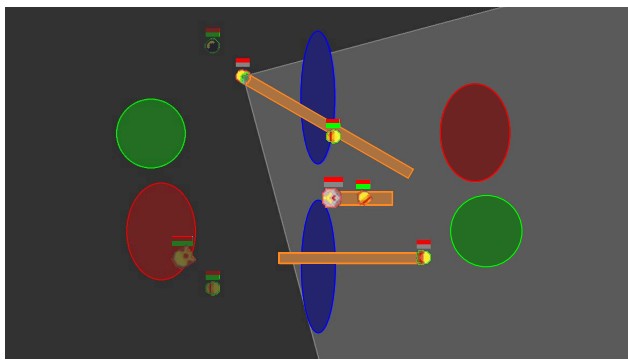

Figure 1. An illustrative scenario showcasing core features of TABX: (a) fan-shaped partial observability, (b) non-targeted interactions, (c) heterogeneous unit roles, and (d) terrain zones that impose complex strategic demands.

et al., 2020; Bard et al., 2020; Ellis et al., 2023; Koyamada et al., 2023). These benchmarks capture key difficulties of multi-agent reinforcement learning (MARL), including partial observability, long-horizon decision-making, high exploration demands, and the need for effective coordination. As a result, they have driven substantial progress in MARL algorithms (Rashid et al., 2020; Kuba et al., 2021; Yu et al., 2022; Gallici et al., 2024).

Despite these advances, most existing MARL benchmarks are built around static task designs. These benchmarks often constrain researchers to predefined scenarios or require extensive codebase modifications to develop new tasks (Kurach et al., 2020; Rutherford et al., 2024b; Matthews et al., 2024a). In practice, a systematic evaluation of MARL algorithms across diverse benchmark conditions and research questions becomes prohibitively costly, as it requires maintaining multiple, often incompatible, environment implementations. Consequently, empirical evaluations often tend toward specific environment configurations, making it difficult to reliably analyze the strengths and weaknesses of different MARL algorithms across diverse conditions (Agarwal et al., 2021). This limitation necessitates the development of frameworks that support scenario customization with parameterized configurations for systematic analysis.

We introduce *Totally Accelerated Battle Simulator in JAX (TABX)*, a scalable, flexible, and easily configurable multi-agent environment designed to evaluate the capabilities of

*Table 1.* Comparison of other RL benchmarks with TABX.

| BENCHMARK | H/W-ACCEL. | MULTI-AGENT | CONFIGURABLE PARAMS | SCENARIO AUTHORING |
|---|---|---|---|---|
| BIPEDALWALKER (WANG ET AL., 2019) | ✗ | ✗ | △ | CODE-LEVEL |
| MINIGRID (NIKULIN ET AL., 2024) | ✓ | ✗ | △ | LAYOUT-BASED |
| CRAFTAX (MATTHEWS ET AL., 2024A) | ✓ | ✗ | △ | CODE-LEVEL |
| KINETIX (MATTHEWS ET AL., 2024B) | ✓ | ✗ | ✓ | CONFIG FILE + GUI |
| GRF (KURACH ET AL., 2020) | ✗ | ✓ | ✗ | CODE-LEVEL |
| SMAC (ELLIS ET AL., 2023) | ✗ | ✓ | △ | CODE-LEVEL |
| JaxMARL (RUTHERFORD ET AL., 2024B) | ✓ | ✓ | △ | CODE-LEVEL |
| JaxNav (RUTHERFORD ET AL., 2024A) | ✓ | ✓ | △ | CODE-LEVEL |
| OGC (RUHDORFER ET AL., 2025) | ✓ | ✓ | △ | LAYOUT-BASED |
| TABX | ✓ | ✓ | ✓ | CONFIG FILE + GUI |

diverse MARL algorithms in rich, structured battle scenarios. Unlike existing battle-based benchmarks (Samvelyan et al., 2019; Ellis et al., 2023; Rutherford et al., 2024b), TABX facilitates rigorous evaluation across diverse strategic landscapes by offering reconfigurable environmental elements. These include parameterized heuristic agents, heterogeneous terrain (e.g., lava, swamp, and bush), and adjustable physical dynamics, allowing researchers to instantiate environments with tailored complexity. Figure 1 provides an overview of a representative environment instance, illustrating the components of TABX.

All of these components are accessible through a configuration interface, enabling users to easily modify and share custom configurations without altering the underlying simulation code. To further support scenario designs, we provide a graphical user interface for designing hand-crafted scenarios via an intuitive visual workflow. Leveraging this high degree of configurability, we demonstrate that controlled variation in various environmental parameters—made practical at scale by TABX's JAX-based, end-to-end vectorized implementation—enables systematic investigation of fundamental MARL challenges. In summary, this paper makes the following contributions:

- We introduce TABX, a high-throughput, JAX-based multi-agent battle simulator with hardware-accelerated execution and large-scale parallel environment sampling, and provide a suite of representative baseline implementations to facilitate standardized benchmarking and reproducible research.

- We propose a customizable framework that supports dynamic reconfiguration of units, terrain, heuristic policies, and physical dynamics without code modification, together with a graphical scenario editor and a collection of predefined scenarios that enable flexible and diverse MARL experimentation.

- We demonstrate that TABX enables systematic analysis of core MARL challenges, including information-dependent value learning, sparse-reward exploration in long-horizon tasks, and zero-shot generalization.

## 2. Related Work

### 2.1. MARL Environments

A diverse array of complex benchmarks have been established to evaluate scalability, coordination, and emergent collaboration in multi-agent reinforcement learning (MARL) (Lowe et al., 2017; Samvelyan et al., 2019; Lanctot et al., 2019; Carroll et al., 2019; Kurach et al., 2020; Ellis et al., 2023; Terry et al., 2021; Rutherford et al., 2024b). While these benchmarks have driven significant progress, their reliance on fixed and static scenario sets constrains the rigorous assessment of MARL algorithms across diverse conditions. The overhead of evaluating algorithms across the disparate conditions of multiple benchmarks remains prohibitively costly, as it necessitates the maintenance of multiple and often incompatible environment implementations. Although certain frameworks permit configuration adjustments, they often rely on cumbersome manual overrides or computationally expensive code-side interventions that disrupt the experiment workflow. Beyond configurability, TABX also differs from existing battle simulators in its underlying mechanics: a direct comparison with SMAX (Rutherford et al., 2024b) on matched scenarios reveals qualitatively different learning dynamics (Section D).

### 2.2. Configurable Environments for Multiple Problems

Recently, diverse problem settings—such as multi-task learning (He et al., 2024), and generalization to unseen tasks (Rutherford et al., 2024a)—have emerged as central challenges in reinforcement learning. Accordingly, various benchmarks have been developed to evaluate these capabilities (Wang et al., 2019; Cobbe et al., 2020; Touati & Ollivier, 2021; Chevalier-Boisvert et al., 2023). However, many of these environments suffer from slow execution speeds, limiting large-scale evaluation across multiple tasks and experimental conditions. Even recent, procedurally rich and GPU-accelerated benchmarks are typically single-agent environments, such as Craftax and Kinetix (Matthews et al., 2024a;b; Bortkiewicz et al., 2025).

While some MARL benchmarks (Zheng et al., 2018; Ruther-

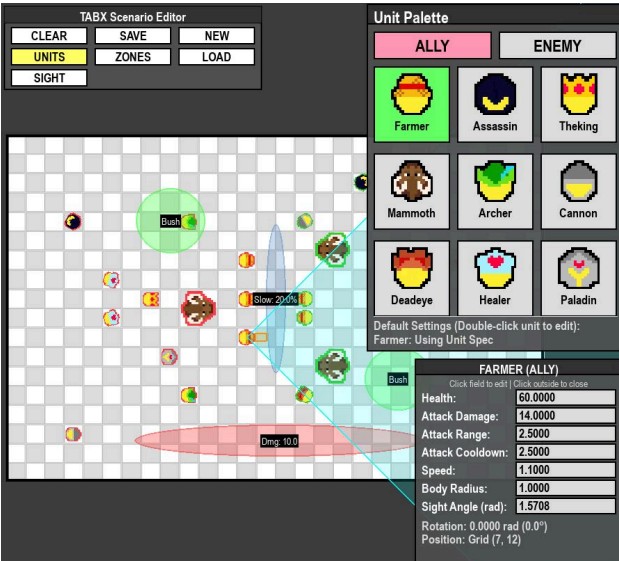

*Figure 2.* Overview of the TABX scenario editor. The interface enables visual authoring of scenarios by allowing users to place ally and enemy units, configure unit specifications, and define environmental zones with adjustable functional effects. The editor provides direct access to key environment parameters through an interactive, code-free workflow.

ford et al., 2024a; Ruhdorfer et al., 2025) offer parameterized interfaces for varying physical task layouts, their configuration space is typically limited to a narrow set of predefined options, particularly regarding terrain components (e.g., layout variations). As a result, they constrain the systematic analysis of agent behaviors across diverse interaction regimes and limit the scope of systematic behavioral analysis by restricting agents to largely homogeneous configurations. In contrast, TABX exposes a high-dimensional parameter space, encompassing physical environmental features alongside unit attributes (e.g., movement velocities and attack ranges).

## 3. Background

**Policy Training and Execution Regimes**   Multi-agent reinforcement learning (MARL) algorithms are typically categorized by the information available during policy training and execution (Albrecht et al., 2024). Most commonly, agents are evaluated in a decentralized manner, where policies are conditioned strictly on local observation histories. Under the *Decentralized Training and Execution* paradigm (e.g., Independent learning), both training and execution are distributed. While this approach scales effectively by avoiding the combinatorial explosion of the joint action space, it suffers from environmental non-stationarity and the inability of agents to leverage the behaviors of others. Conversely, the *Centralized Training and Decentralized Execution (CTDE)* framework aims to combine the benefits of both centralized training and decentralized execution.

By utilizing a centralized value or global state information during training, CTDE algorithms enable agents to learn coordinated behaviors in a computationally tractable manner while maintaining decentralized execution during inference.

**Unsupervised Environment Design**   Unsupervised Environment Design (UED) (Dennis et al., 2020) has proven effective in enhancing generalization both within and beyond an environment's initial contextual MDP space (Modi et al., 2018). UED generalizes the training process as a game between a student agent and an adversary that adaptively selects environment parameters, denoted as levels. The adversary's objective is to maximize the agent's regret, defined as the gap between the return of the optimal policy and that of the current agent $\pi_i$ for a given level $\theta$. This adversarial mechanism induces an emergent curriculum of increasing complexity, which has been shown to significantly improve both learning efficiency and the zero-shot generalization capabilities of the resulting policy.

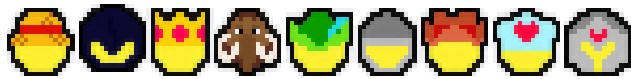

## 4. Totally Accelerated Battle Simulator in JAX

We propose *Totally Accelerated Battle Simulator in JAX (TABX)*, a rapid, flexible, and easily configurable sandbox for MARL. Similar to established benchmarks (Samvelyan et al., 2019; Berner et al., 2019; Ellis et al., 2023; Rutherford et al., 2024b), TABX requires a team of agents to engage in combat against opposing forces. TABX incorporates additional environmental primitives—such as fan-shaped fields of view, non-targeting interaction mechanism, various terrain zones, and parameterized heuristic agents—that significantly augment the strategic complexity of the task. We provide further details of TABX in Section A.

Each agent possesses a partial, fan-shaped observation field oriented along its facing direction, analogous to a first-person perspective, illustrated as the gray region in Figure 1. Within this field, agents observe the current status (e.g., remaining health points), specifications of all visible units, and terrain zone information. As a result, agents must actively rotate and navigate the environment to detect and engage enemies effectively. This constrained observation model complicates coordinated maneuverability, requiring agents to jointly optimize positioning and heading to maintain situational awareness. Conditioned on their observations, agents select from seven discrete actions: four directional movements, an attack action (supporters use a heal action instead), and two rotations (left and right). The attack action can only be triggered once the unit's cooldown is up.

A distinguishing aspect of TABX is its unit interaction system, which incorporates non-targeted attack and healing

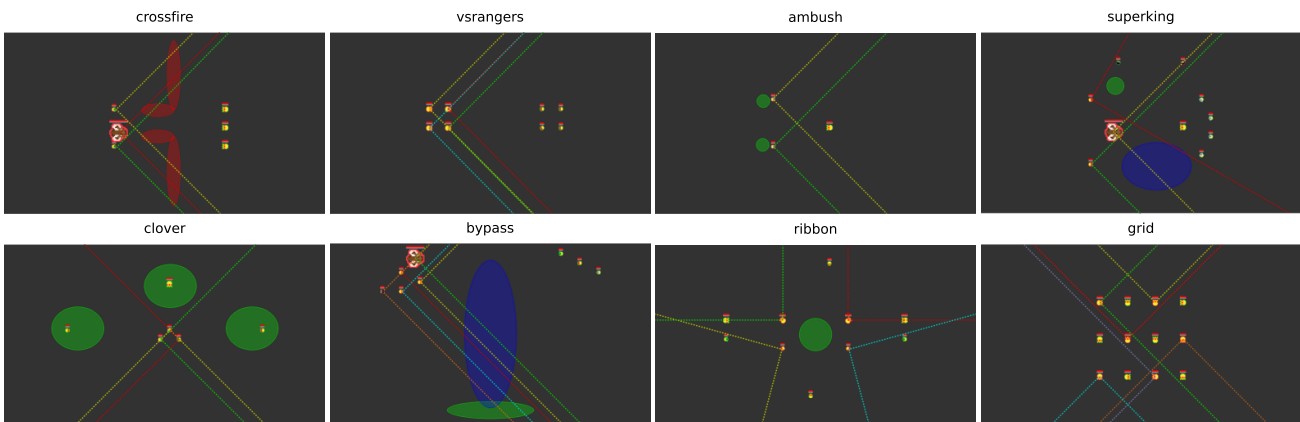

*Figure 3.* Representative designed scenarios illustrating different degrees of dependence on global state information. Colored dashed lines indicate the fan-shaped fields of view of individual allies, highlighting how partial observability and viewpoint separation vary across scenarios. Colored ellipses represent terrain zones with distinct functional effects, such as movement speed reduction or visibility occlusions. Allies are denoted by a red outline, while enemies are indicated by a green outline.

mechanisms. Unlike many existing battle simulators that rely on explicit target selection, units in TABX execute interactions within a hurtbox, a spatial interaction region aligned with their current heading. As illustrated in Figure 1, units positioned within this region are affected by the interaction: for offensive units, enemy units receive damage, whereas for healer units, allied units receive healing. An interaction occurs only when a unit is positioned within the hurtbox at the time of execution. Comprehensive specifications of the non-targeting mechanics are provided in Section A.3.

**Units and Zones**    TABX provides a total of nine distinct units, comprising four melee units, three ranged units, and two supporter units. Each unit is characterized by a vector of intrinsic specifications, encompassing both physical attributes and combat capabilities (attack or healing). In addition, TABX supports terrain zones with specialized functional effects, such as periodic damage in lava, concealment in bushes, and velocity attenuation in swamps. Bush zones, in particular, induce asymmetric visibility; a unit within a bush is hidden from opponents unless it shares the same bush, though it remains observable to its own teammates. Furthermore, if a unit in a bush engages in combat—either by attacking or being attacked—it becomes temporally revealed to the opposing team. This mechanism introduces a layer of partial observability that significantly increases the strategic complexity of adversarial engagements. Further details of zones are provided in Section A.7.

### 4.1. Environmental Parameters

TABX offers a diverse set of environmental parameters that enable custom configurations to address specific research questions, such as evaluating the fundamental properties and generalization of various MARL algorithms. These pa-

rameters span four primary dimensions of the environment: unit specifications, environmental zones, heuristic policy parameters, and physical dynamics. In TABX, these parameters are dynamically reconfigurable, allowing environment conditions to be varied across episodes without code modification or recompilation. This capability is essential for systematic studies that require controlled variation over task distributions—such as analyzing sensitivity to environmental factors, evaluating robustness and generalization, or applying unsupervised environment design—while preserving JAX-based GPU efficiency and end-to-end vectorized execution.

Each category governs distinct environmental facets: unit parameters define agent attributes (e.g., health points and attack damages) for both allies and enemies; terrain parameters instantiate zones with specialized functional effects (e.g, asymmetric concealment and movement speed penalty); heuristic policy parameters govern the proficiency of non-player unit behaviors; and physics parameters modulate the underlying simulation dynamics, including the temporal step size, collision tolerances, and coefficients of restitution. Further details are provided in Section A.5.

**Graphical Scenario Editor**    To facilitate flexible scenario design, TABX provides a graphical user interface (GUI) that allows users to construct custom scenarios and their parameters through an intuitive visual workflow. Figure 2 illustrates the TABX scenario editor, which exposes these configuration dimensions through an interactive graphical interface. Using the editor, users can visually compose scenarios by placing units and terrain zones, adjusting unit specifications, and modifying zone effects without altering the underlying simulation code. We provide comprehensive specifications for all predefined scenarios and a guide to the GUI editor in Section C.

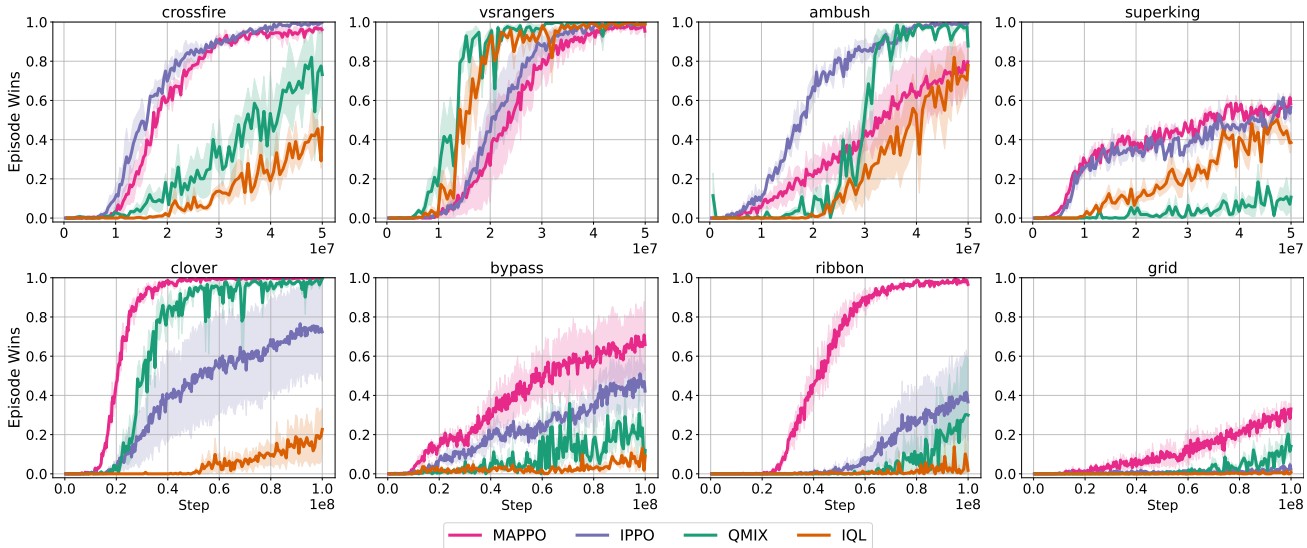

*Figure 4.* Average episode win rates for baseline algorithms across eight different scenarios.

## 4.2. Role-Appropriate Heuristic Policy

While prior benchmarks such as SMAX and MPE (Rutherford et al., 2024b) support heuristic policies for adversaries to shift the focus from competitive scenarios to cooperative tasks, these policies are typically simplistic and homogeneous. Consequently, they fail to represent the strategic diversity inherent in complex multi-agent interactions, limiting their effectiveness as evaluation tools. To address these limitations, we propose a role-appropriate heuristic policy wherein each unit attribute contributes an independent behavioral bias. The final action of a unit emerges from the composition of these role-specific primitives, allowing for the generation of diverse and sophisticated adversarial behaviors. We provide several levels of expertise for the heuristic policy (e.g., random, novice, and medium), categorized by specific parameter thresholds in Section A.5.

To facilitate systematic classification and support the design of heuristic agents, we define three orthogonal functional attributes: Assassin, Ranger, and Healer. These roles are programmatically derived from a unit's underlying kinematic and combat parameters (e.g., movement speed). Further details regarding functional attributes and their thresholds are provided in Section A.5 and Section A.8, respectively.

Due to the fan-shaped field of view and the non-targeting interaction model in TABX, heuristic agents adopt a "seek-and-approach" strategy as their default behavior. Role-specific attributes then modulate this baseline by shaping how agents position themselves during engagements. Specifically, Assassin units favor flanking maneuvers against vulnerable opponents, Ranger units regulate distance to maintain range advantages and exploit bush-induced occlusion, and Healer units prioritize proximity to injured allies to provide sustained support.

## 5. Experiments

Our experiments are organized around scenarios tailored to specific research questions. First, we investigate scenarios that exacerbate disjoint partial observability (Section 5.3). Second, we examine exploration challenges within long-horizon scenarios characterized by sparse reward signals (Section 5.4). Finally, we conduct a zero-shot generalization study by applying Unsupervised Environment Design (UED) to a MARL algorithm (Section 5.5)—a process facilitated by TABX's ability to reset environmental parameters without requiring JIT re-compilation (e.g., dynamically varying unit specifications or zone layouts across episodes).

Each scenario is constructed by first identifying a target MARL challenge and then designing unit compositions and zone placements to isolate that specific variable. For instance, the clover scenario is intended to demonstrate the necessity of global information: enemies remain occluded or positioned beyond the agents' immediate perceptual fields during the initial timesteps, with bushes placed to hide enemies and enhance partial observability, while the ally team is composed of ranged units that can scout the bushes from a distance. Comprehensive scenario specifications are provided in Section C.1.

### 5.1. Baselines and Settings

We evaluate a suite of standard baselines across diverse scenarios. For multi-agent reinforcement learning (MARL), we employ Independent PPO (IPPO) (De Witt et al., 2020), MAPPO (Yu et al., 2022), Independent Q-Learning (IQL) (Tampuu et al., 2017), and QMIX (Rashid et al., 2020). For the UED framework, we evaluate several algorithms using MAPPO as the base learner: Domain Randomization (DR) (Jakobi, 1997; Sadeghi & Levine, 2016), PLR (Jiang et al.,

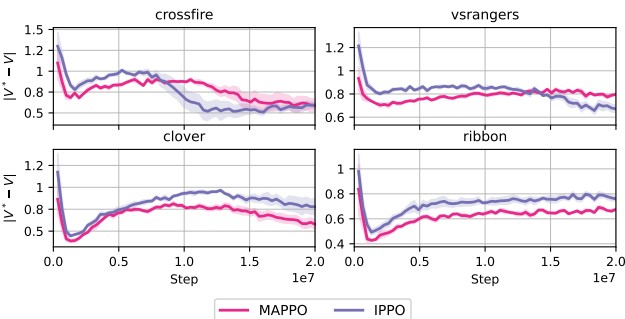

*Figure 5.* Value estimation error $|V - V^*|$ across different scenarios. $V^*$ is approximated via extensive rollout simulations. Results compare centralized value learning (MAPPO) with independent learning (IPPO).

2021a), Robust PLR (PLR$^\perp$) (Jiang et al., 2021b), ACCEL (Parker-Holder et al., 2022), and SFL (Rutherford et al., 2024a). From the options offered by TABX, we primarily focus on two categories of free parameters (unit specifications and zones) and implement a level generator that leverages these parameters to create diverse environments.

In all our experiments, results are averaged over five random seeds, with shaded regions in figures indicating the standard error. For our experimental evaluation, we categorize opponent behavior into discrete difficulty tiers. In the MARL benchmarks, opponent agents are governed by a medium-level heuristic policy to provide a consistent baseline for performance comparison. Conversely, within the UED framework, we initialize opponents with a novice-level heuristic for unit- and zone-variation experiments, focusing on analyzing the robustness of the agent to scenario variations. The implementations used in our experiments are made available to support reproducibility. We provide comprehensive experimental settings, implementation details, and hyperparameter configurations in Section E.

## 5.2. Benchmarks of MARL Algorithms

We evaluate the baselines across various predefined scenarios. As shown in Figure 4, MAPPO and IPPO achieve strong performance across most scenarios, but their relative performance varies depending on the properties of the scenario. In particular, IPPO outperforms MAPPO in `crossfire`, `vsrangers`, and `ambush`, indicating that the advantages of centralized value learning are not uniform and can depend critically on the structure of partial observability and state information available during training. This observation is consistent with prior findings that centralized critics may suffer from poor generalization or misleading value estimates in certain settings, where independent learning can be more effective (De Witt et al., 2020).

Among value-based methods, QMIX performs strongly than IQL across multiple scenarios. This behavior is consistent with QMIX's inductive bias toward cooperative value decomposition induced by its monotonic mixing network, which enforces a structured relationship between individual and joint value functions aligned with the Individual-Global-Maximum (IGM) principle (Son et al., 2019) and can facilitate coordination under shared reward structures. The `ribbon` and `grid` scenarios pose significant challenges, reflecting the increased difficulty induced by disjoint partial observability and complex spatial interactions.

Overall, scenarios in TABX exhibit significant sample complexity, requiring more extensive training to develop effective combat strategies. This increased difficulty is driven by TABX's distinctive features, which necessitate more nuanced coordination and exploration. In addition to episode win rate, we report several complementary metrics—including first-kill rate, total episode length, and episode returns. These results are provided in Section J.

## 5.3. Centralized Value Learning

TABX enables a controlled analysis of centralized value learning by allowing us to systematically vary the extent to which accurate value estimation depends on global state information. Leveraging this flexibility, we construct a set of tasks that explicitly differentiate between scenarios where centralized information is largely redundant and those where it is essential. Representative examples of these scenarios are visualized in Figure 3, which illustrates varying degrees of dependence on global state information.

In scenarios where accurate value estimation can be achieved through local observations alone, such as `crossfire` and `vsrangers`, the additional global information available to a centralized value function offers no inherent structural advantage over independent training. Consequently, these scenarios serve as controlled evaluation settings for centralized critics, exposing failure modes arising from reliance on irrelevant or redundant global features.

In contrast, the `clover` and `ribbon` scenarios demand long-horizon reasoning under partial observability, as agents must account for enemies that remain occluded or positioned beyond their immediate perceptual fields during the initial timesteps. In these environments, accurate value estimation fundamentally depends on global state information that is inaccessible to individual agents but available to the centralized critic during training. These scenarios therefore represent favorable conditions for centralized value learning, enabling a focused analysis of its potential advantages beyond raw policy performance.

To quantify these effects, we compute the value estimation error between the learned value function $V$ and a Monte Carlo estimate of the true value under the current policy $V^*$, obtained via extensive rollout simulations (see Section F.3 for details). As shown in Figure 5, the results exhibit a

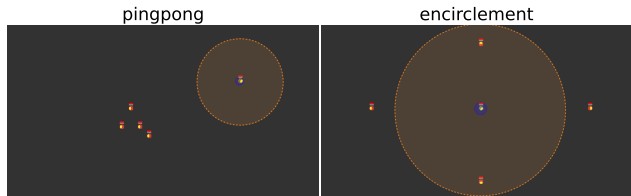

*Figure 6.* Illustration of exploration scenarios `pingpong` and `encirclement`. The orange dashed circle denotes the attack range of the stationary enemy unit.

clear scenario-dependent pattern: centralized value learning achieves substantially lower estimation error in scenarios that require global information, while offering limited or no advantage when local observations are sufficient. Although the absolute magnitude of the error gap may appear modest, it is significant relative to the effective range of the discounted return, since the win/loss reward is heavily discounted across long episodes; a correlation analysis (Section G) yields a strong negative correlation between value error and episode return (Pearson $r = -0.82$), confirming that even modest reductions in value estimation error translate into meaningful performance gains. These findings underscore the importance of aligning the structure of centralized critics with the informational demands of the environment, consistent with the performance trends observed in Figure 4.

## 5.4. Exploration

As shown in Figure 6, `pingpong` and `encirclement` are designed as long-horizon tasks with sparse rewards, where a positive signal is difficult to obtain due to the high precision required to land a single hit on the enemy. Entering the enemy's attack range incurs sustained negative rewards until an attack hits the enemy. As a result, agents must traverse a region of consistently unfavorable returns before reaching a rewarding state. This reward structure makes naive exploration ineffective, as success depends on engagement behaviors through high-level exploration.

To highlight the resulting exploration challenges, we incorporate Random Network Distillation (RND) (Burda et al., 2019) as a representative baseline to address the exploration challenges inherent in our environment. We integrate RND into the MAPPO framework by generating an intrinsic reward based on the global state representation. This approach provides an additional exploration bonus that encourages the agents to discover novel global states. Detailed specifications of the RND implementation are provided in Section F.

Figure 7 illustrates the average episode win rates for MAPPO across various entropy coefficients and with the addition of RND. While RND demonstrates significant performance gains in both scenarios, MAPPO with a higher entropy coefficient ($\sigma = 0.01$) also performs competitively

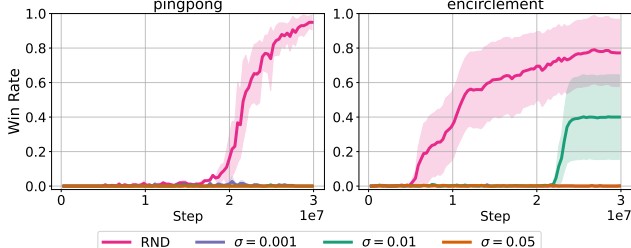

*Figure 7.* Average episode win rates for MAPPO across different entropy coefficients ($\sigma = \{0.001, 0.01, 0.05\}$) and RND in exploration scenarios.

in `encirclement`. Nevertheless, given that both tasks require high-level exploration to overcome initial sparse rewards, suggesting that efficient exploration mechanisms play a critical role in accelerating learning. In `pingpong`, RND achieves meaningful win rates after 15M environment steps, subsequently enabling the agents to converge toward the optimal combat strategy. In `encirclement`, although RND facilitates faster initial progress, it exhibits high variance; MAPPO with $\sigma = 0.01$ reaches comparable win rates after approximately 20M environment steps.

## 5.5. Zero-shot Generalization

We conduct a multi-agent zero-shot generalization study by integrating representative UED algorithms with the MAPPO framework. In our experiments, we utilize each level through two independent categories of free parameters: unit specifications and zone layouts. Unit specification parameters include maximum health, speed, and attack damage, while zone parameters encompass zone type, $xy$-position, ellipse axes sizes, and effect values (e.g., lava damage). These diverse parameters allow us to evaluate generalization across unit attributes and environmental structure.

Using this formulation, we construct four training scenarios by combining diverse unit compositions with six environmental zones. For each scenario, agents are trained on levels sampled uniformly from the corresponding parameter spaces. We then evaluate zero-shot generalization for each free parameter type: four unit-specification scenarios and three zone scenarios, allowing us to assess how well policies transfer to unseen agent attributes and environmental layouts. We additionally evaluate zero-shot generalization across a spectrum of heuristic opponent tiers, which serves as a complementary axis of variation. Results and detailed setup are reported in Section J.5 and Section F.

Figure 8 reports the average episode win rates for UED baselines evaluated under different types of parameter variation. Across all settings, agent performance improves consistently throughout training, but the resulting performance profiles differ markedly depending on whether variation is introduced through terrain layouts or unit specifications. Interestingly, while replay level-based methods exhibit comparable

*Figure 8.* Average zero-shot win rates of UED baselines aggregated across evaluation scenarios. The two leftmost panels show generalization to unseen unit specifications and zone layouts; the three rightmost panels isolate individual unit attributes (health, speed, attack damage) one at a time during training.

performance across both free parameter categories, they significantly underperform relative to DR in unit-specification evaluations. One potential explanation is that unit specifications are intrinsically linked to the agents' controllable dynamics; consequently, exposure to a diverse range of levels enhances the agents' robustness and generalization to unseen configurations.

Although most methods demonstrate significant performance gains in terrain zone-variant tasks, all evaluated baselines struggle to generalize to unseen unit specifications, finally yielding average win rates below 50%. We attribute this gap to the substantially higher dimensionality and wider effective range of the unit-specification parameter space, which jointly modulates multiple attributes (e.g., health, speed, attack damage) across every controllable agent rather than zones. UED methods prioritize environment parameters with the highest estimated learning progress, naturally expanding coverage over diverse configurations. However, such high-dimensional configuration spaces are characteristic of multi-agent UED settings, where the number of free parameters scales with the number of agents. These spaces interact poorly with the adaptive curricula used by existing UED methods, misaligning with what the agent can effectively learn and thereby destabilizing training. We further conduct isolated ablations over individual unit attributes. As shown in the rightmost three panels of Figure 8, agents trained with only health variation achieve substantially lower win rates than those trained with speed or attack damage variation alone, identifying health as the dominant bottleneck for generalization.. Health induces a substantially wider effective difficulty range than speed or attack damage, creating a broad regret landscape that interacts poorly with UED's adaptive curriculum.

These results underscore that generalization induced by changes in unit attributes presents a qualitatively distinct challenge compared to spatial generalization, which is the primary focus of existing benchmarks (Chevalier-Boisvert et al., 2023; Ruhdorfer et al., 2025). In particular, variations in unit specifications directly affect the agents' controllable dynamics, requiring exposure to a broader distribution of environment configurations in order to achieve

robustness to unseen attributes. By offering independently tunable zone- and unit-level parameters, TABX provides a controlled testbed for diagnosing the limitations of MARL UED algorithms under varying parameter dimensionality and range. More broadly, our findings suggest that scaling UED to high-dimensional, agent-level parameter spaces remains an open challenge for multi-agent curriculum learning.

### 5.6. Scalability of TABX

To assess the scalability of our environment, we conduct experiments measuring throughput as a function of the number of parallel environments executed on a single GPU (NVIDIA RTX 4090) and on a multi-core CPU (AMD Ryzen Threadripper PRO 7975WX, 32 cores).

As shown in Figure 9, TABX exhibits near-log-linear throughput scaling with respect to the number of parallel environments across a wide range of unit counts. While increasing the number of units per environment reduces absolute throughput, the scaling trend remains consistent, indicating that TABX effectively exploits hardware parallelism and maintains stable performance as both environment and agent complexity grow.

We next compare TABX against SMAX, which requires re-compilation if it is instantiated with a new scenario under comparable conditions. For this comparison, we report the effective environment steps per second across 100 distinct, randomly generated scenario resets, explicitly including the JIT-compilation overhead incurred at each reset. This evaluation protocol reflects the end-to-end latency experienced during training when environments are frequently reconfigured with new scenarios. As summarized in Figure 10, TABX consistently achieves substantially higher throughput than SMAX, with performance that is largely invariant to the number of units. These results highlight the practical advantage of TABX for large-scale training settings that require frequent scenario reconfiguration.

To verify that this advantage stems from the elimination of recompilation overhead rather than from inherently faster per-step dynamics, we additionally conduct a fixed-

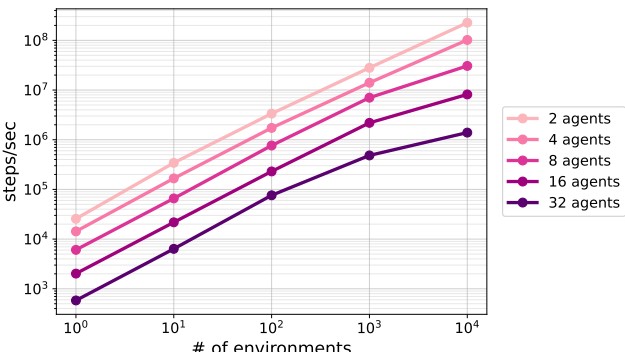

*Figure 9.* Scalability of TABX with increasing numbers of parallel environments. Lines correspond to different numbers of units per environment, illustrating how throughput scales with both environment and unit count.

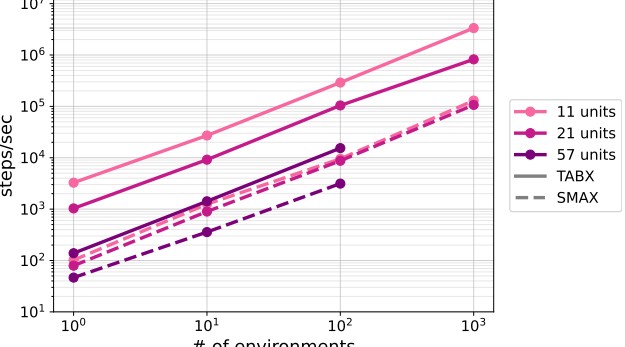

*Figure 10.* Speed comparison between TABX and SMAX with an increasing number of parallel environments. Results represent the sampling rate across 100 distinct scenarios, explicitly including JIT compilation overhead, using a random policy for interaction.

configuration comparison in which no scenario resets occur during measurement (See Section H). In this regime, TABX can be slower than SMAX for smaller numbers of parallel environments and larger team sizes, reflecting the higher per-step cost of its richer dynamics (fan-shaped field of view, hurtbox resolution, terrain effects) relative to SMAX's distance-based checks. This confirms that TABX's throughput advantage in the reconfiguration setting arises from its recompilation-free design rather than from a coarser action space or simpler simulation.

### 5.7. Competitive Evaluation

Beyond heuristic opponents, TABX also supports policy-vs-policy evaluation, where independently trained policies compete directly against each other. This follows naturally from the environment API, which accepts actions for both teams and returns team-wise rewards, allowing learned policies to replace heuristic opponents without modifying the environment.

To demonstrate this capability, we train policies using MAPPO, IPPO, QMIX, and IQL for both the ally and enemy sides, then evaluate pairwise matchups across scenarios. As shown in Section I, MAPPO achieves the strongest average performance across opponents, consistent with its advantage in scenarios that benefit from centralized value learning. This setup enables competitive comparisons between MARL algorithms and supports future research on self-play, opponent modeling, and population-based training.

## 6. Limitation

TABX is intentionally designed as a combat-oriented benchmark, and therefore does not cover cooperative MARL settings such as resource sharing, task allocation, or communication emergence. While the experiments in this work demonstrate the environment's capabilities, they are primarily illustrative and do not yet yield fundamentally new insights into MARL algorithms. A promising future direction is transfer learning, where agents first acquire low-level control skills in single-agent settings before being deployed in multi-agent environments. This could help disentangle the challenges of individual control from those of multi-agent cooperation. More broadly, hierarchical decomposition of cooperative tasks is a promising direction for future research with TABX.

## 7. Conclusion

We introduce TABX, a scalable and highly configurable multi-agent environment for evaluating MARL algorithms in structured battle scenarios. By exposing unit specifications, terrain zones, heuristic behaviors, and physical dynamics as primary parameters, TABX enables systematic environment construction tailored to diverse research questions while maintaining JAX-based GPU efficiency. Beyond the specific studies presented in this work, TABX supports the principled design of configurable multi-agent scenarios. By leveraging this configurability, we demonstrate how controlled variation over environment structure and unit specification enables systematic investigation of core challenges such as information-dependent value learning, exploration in long-horizon tasks, and generalization to unseen configurations. Overall, TABX provides an efficient testbed for advancing research on the interaction between environment design and MARL research.

While TABX provides a robust functional foundation, opportunities remain to further increase strategic complexity. Future work will extend TABX's components by introducing terrain obstacles that restrict unit line-of-sight, incorporating new structural elements such as fortifications and defensive walls to support novel task types, supporting representation learning research by offering pixel-based observations, and integrating active unit-specific skills that transcend static interaction variations.

## Impact Statement

TABX is intended as a research testbed for multi-agent reinforcement learning, supporting the study of cooperation, exploration, and generalization. Following precedent in prior MARL benchmarks (e.g., SMAC, SMACv2, JaxMARL), it adopts a stylized combat framing because such tasks naturally induce rich multi-agent dynamics, but we explicitly do not intend TABX as a model of any real-world combat or operational context—the simulator's abstractions (fixed unit archetypes, idealized physics, small-scale tactical scenarios) bear no resemblance to real military systems, and we discourage interpreting TABX results as evidence relevant to military capability or deployment. By lowering the barrier for systematic MARL research—particularly on algorithmic robustness and generalization—we hope TABX contributes to developing multi-agent algorithms whose behavior is well-understood across diverse conditions, a prerequisite for responsible deployment in any applied domain.

## Acknowledgement

This work was partly supported by Institute of Information & Communications Technology Planning & Evaluation (IITP) grant funded by the Korea government (MSIT) (No. RS-2022-II220311, Development of Goal-Oriented Reinforcement Learning Techniques for Contact-Rich Robotic Manipulation of Everyday Objects (31%), No. RS-2024-00457882, AI Research Hub Project, No. RS-2019-II190079, Artificial Intelligence Graduate School Program (Korea University), the IITP (Institute of Information & Communications Technology Planning & Evaluation)-ITRC (Information Technology Research Center) grant funded by the Korea government (Ministry of Science and ICT) (IITP-2026-RS-2024-00436857) (31%), the NRF (RS-2024-00451162) funded by the Ministry of Science and ICT, Korea, BK21 Four project of the National Research Foundation of Korea, the National Research Foundation of Korea (NRF) grant funded by the Korea government (MSIT) (RS-2025-00560367), the IITP under the Artificial Intelligence Star Fellowship support program to nurture the best talents (IITP-2026-RS-2025-02304828) grant funded by the Korea government (MSIT) (32%), and KOREA HYDRO & NUCLEAR POWER CO., LTD (No. 2024-Tech-09)

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

# A. Details on TABX

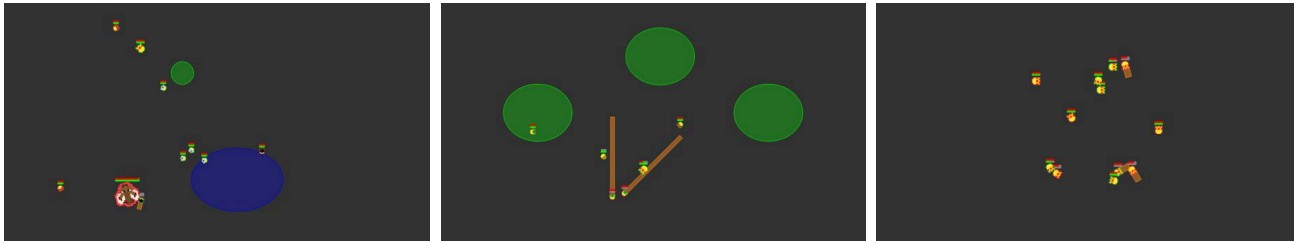

Our code is available at: https://github.com/ku-dmlab/TABX.

## A.1. Observation and Global State

Each agent receives a structured observation composed of three components: (i) its own local state, (ii) information about other units within its observable region, and (iii) environment-level zone features. Observations are constructed under partial observability constraints imposed by each unit's field of view and environmental effects.

**Own-unit features.**   The observation of each agent always includes its own state variables, independent of visibility. These features encode the unit's current condition and physical properties, including health and normalized maximum health, absolute position, orientation, attack range, attack damage, attack cooldown and normalized cooldown ratio, body radius, body mass, sight angle, alive status, and current movement speed.

**Other-unit features.**   For all other units in the environment, each agent observes a set of relative features conditioned on visibility. Observed features include the target unit's health and normalized maximum health, relative position, orientation, attack range, attack damage, cooldown and normalized cooldown ratio, body radius, body mass, sight angle, alive status, team affiliation, attackability flag, and movement speed. All other-unit features are masked by a visibility matrix derived from field-of-view constraints and zone-specific rules (e.g., bush zones), such that non-visible units contribute zero-valued features.

**Zone features.**   If environmental zones are present, each agent additionally observes zone-specific information. For each zone, the observation includes the zone type, relative position with respect to the observing agent, geometric parameters of the zone, and its associated effect magnitude. Zone features are masked when the zone type indicates inactivity, ensuring a fixed observation dimensionality across different scenarios.

**Global state.**   In addition to per-agent observations, the environment provides a global state representation for centralized training. The global state consists of the full set of unit-level features for all agents, augmented with zone-level features when applicable. This representation provides complete environment information without visibility masking and is used exclusively for training-time components such as centralized critics.

## A.2. Action Space

The action space $\mathcal{A}$ is discrete and consists of four primary components: cardinal movement, rotation, an attack, and a no-op (idle) action. Movement actions are defined in a global (world) coordinate frame and are executed independently of the agent's current orientation. Notably, the movement actions are defined within a global coordinate frame, where directional commands are executed independently of the agent's current orientation. The environment provides a state-dependent action mask, which dynamically restricts the available action space. Specifically, the attack action is masked as invalid during its cooldown period, preventing the agent from issuing illegal commands before the internal timer has elapsed.

## A.3. Interaction Dynamics

We model agent interactions through (i) discrete action execution, (ii) continuous-time physics integration, and (iii) event-driven combat and visibility updates. Each environment step proceeds in three stages: a physics update for all dynamic entities, an action application stage (movement/rotation/combat), and a post-processing stage that applies zone effects and

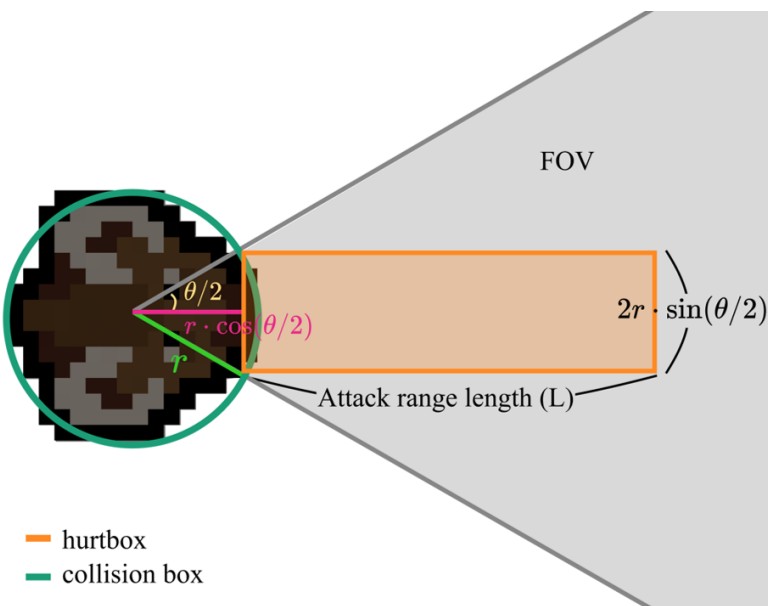

*Figure 11.* Each unit is associated with a forward-facing rectangular hurtbox of length $L$, corresponding to its attack range, and bounded laterally by the field of view (FoV). An attack is registered whenever a target unit's circular body collider intersects the hurtbox.

updates episode termination conditions. Disabled units correspond to inactive placeholders used for scenario padding (e.g., when the number of spawned units is smaller than the maximum supported unit count) and do not participate in decision making or rewards.

**Movement and kinematic update.**  At every timestep, each unit updates its cooldown and advances its physical state via explicit Euler integration: velocities are updated from the current acceleration and positions are then advanced using timestep $\Delta t$. To discourage leaving the playable area, we impose a boundary penalty: if a unit's position violates the arena bounds, it receives continuous damage of $0.1\,h_{\max}\Delta t$ per step, where $h_{\max}$ denotes the unit's maximum health. The position is subsequently clipped to the valid region.

**Collision detection and resolution.**  After the kinematic update, collisions are evaluated pairwise among all units equipped with circular colliders. For each interacting pair, we compute the contact normal $\mathbf{n}$ and penetration depth $d$ using a circle-to-circle closest-point formulation.

When $d > 0$, collisions are resolved via a mass-aware impulse response with restitution $e$, followed by positional correction to eliminate interpenetration. The impulse magnitude is computed from the relative velocity projected onto the contact normal, and the resulting impulse is distributed according to the inverse masses of the two units. To improve numerical stability, we apply a penetration slop threshold $\varepsilon$ and a fractional correction factor $\alpha$ (percent), ensuring that only penetrations exceeding $\varepsilon$ contribute to position correction.

If the relative normal velocity indicates that the objects are already separating, i.e., $\mathbf{v}_{\mathrm{rel}} \cdot \mathbf{n} > 0$, only positional overlap correction is applied without impulse response. Units with kinematic bodies are exempt from impulse and correction updates. Notably, collision detection and resolution are performed regardless of a unit's alive status; units with zero health are still considered during collision handling to preserve consistent physical interactions.

**Attack range geometry and target selection.**  Combat interactions are governed by a simple geometric attack region defined in front of the attacking unit (Fig. 11). Each unit is associated with a forward-facing rectangular *hurtbox* whose length $L$ corresponds to the unit's individual attack range, aligned with its current orientation and bounded laterally according to its field of view. An attack is considered valid whenever another unit's body collider intersects this rectangular hurtbox.

This formulation allows each unit to strike targets located directly in front of it within a limited angular span and distance determined by its own attack range, closely matching the geometric interpretation shown in Fig. 11. Based on this overlap test, we construct a binary *attackable matrix* indicating which unit pairs can interact. This matrix is further constrained by

unit roles: default attack units may only target enemies, while healing-type units may only affect allies. Among all valid targets within the hurtbox, each attacker selects the closest one as the attack target for the current timestep. In parallel, a field-of-view based visibility mask is applied, ensuring that only units located within the forward viewing cone and not concealed by environmental effects (e.g., bushes) are observable to the agent.

**Attack execution and damage application.**    If an agent chooses the combat action and satisfies the cooldown constraint, it becomes eligible to attack in the current step. An attack succeeds if the selected target is marked attackable; successful attacks are recorded symmetrically in an interaction matrix and produce instantaneous damage (or healing) equal to the unit's attack damage parameter. Damage is applied via an event notification mechanism: the victim unit checks whether it is the designated target of an eligible attacker and, if so, reduces its health accordingly with clamping to $[0, \text{max\_health}]$.

**Zone-mediated interaction effects.**    Environmental zones introduce additional interaction dynamics. Lava zones apply continuous damage over time to units inside the zone. Swamp zones apply a multiplicative speed reduction while the unit remains inside. Bush zones implement partial observability with rule-based visibility: units inside bushes are hidden from the opposing team unless they share the same team, occupy the same bush, or reveal themselves through attacking/being attacked, after which visibility information is shared among relevant teammates.

**Post-step bookkeeping.**    physics, actions, and zone effects, each unit's alive status is updated based on whether health remains positive, and disabled units are treated as done. Episode termination is triggered when all units of all but one team are eliminated (or disabled), or when a maximum-horizon truncation condition is met.

### A.4. Reward

Agents receive a shared reward proportional to the marginal difference between the total health ratios of allies and adversaries, incentivizing both offensive engagements and curative actions. Specifically, at each timestep $t$, the shared dense reward is defined as

$$r_t = \Delta \left( \frac{1}{|\mathcal{A}|} \sum_{i \in \mathcal{A}} \frac{h_i^t}{h_i^{\max}} - \frac{1}{|\mathcal{E}|} \sum_{j \in \mathcal{E}} \frac{h_j^t}{h_j^{\max}} \right), \tag{1}$$

where $\mathcal{A}$ and $\mathcal{E}$ denote the sets of allied and adversarial agents, respectively, and $h^t$ and $h^{\max}$ represent the current and maximum health values.

Upon episode termination—triggered either by the incapacitation of all agents on one team or by reaching the maximum step horizon—agents receive a sparse terminal reward of $+1$ for a win and $-1$ for a loss. The negative terminal reward often induces overly conservative behavior, causing agents to avoid combat to minimize potential losses.

In cases of episode truncation (i.e., reaching the maximum step horizon), the win–loss outcome is determined by comparing the final average health ratios of the two teams. If the health ratios are equal, the allied team is declared the loser. This asymmetric tie-breaking rule is intentionally introduced to discourage evasive strategies that rely solely on survival without meaningful engagement.

### A.5. Environment Parameters

The TABX environment is formalized as a parameterized transition function, where the task distribution is governed by a configuration vector passed during the environment reset procedure. This configuration is encapsulated as a structured object comprising four primary axes: unit information (e.g., health, position), spatial zone distributions (zone type, position), heuristic policy coefficients for non-player entities (e.g., random, novice, medium), and physical environment constants (e.g., timestep $\Delta t$, restitution $e$).

To parameterize the opponent's behavioral complexity, we discretize the rule-based policies into a hierarchical set of difficulty tiers. These tiers, summarized in Table 2, define a spectrum of tactical sophistication, enabling us to evaluate agent robustness against varying levels of adversarial competence. The environment dynamics and agent behaviors are governed by a set of continuous parameters. The stochasticity parameter $\epsilon \in [0, 1]$ determines the probability of an agent executing a random action instead of its intended policy. For specialized units, we define tactical primitives: the Ranger unit employs a "kiting" strategy to maintain a spatial standoff from opponents, triggered when the distance falls below an aggressive

threshold $\xi$. Furthermore, unit classifications are determined by their intrinsic attributes; specifically, the movement speed defines the Assassin class, while the effective engagement range defines the Ranger class.

*Table 2.* Heuristic policy parameters across all difficulty tier.

| LEVEL | STOCHASTICITY OF ACTION ($\epsilon$) | AGGRESSIVE THRESHOLD ($\xi$) | ASSASSIN SPEED | RANGER ATTACK RANGE |
|---|---|---|---|---|
| RANDOM | 1.0 | 0.0 | 1.4 | 10.0 |
| NOVICE | 0.5 | 0.1 | 1.4 | 10.0 |
| MEDIUM | 0.2 | 0.3 | 1.4 | 10.0 |
| ADVANCED | 0.1 | 0.5 | 1.4 | 10.0 |
| EXPERT | 0.01 | 0.7 | 1.4 | 10.0 |

## A.6. Unit Specifications

We provide nine predefined units: Farmer (F), Assassin (S), TheKing (K), Mammoth (M), Archer (A), Cannon (C), Deadeye (D), Healer (H), and Paladin (P). Each unit is characterized by multiple specification components, including health, body radius, body weight, speed, attack damage, attack range, attack cooldown, sight angle, and occupied space. Detailed unit statistics are provided in Table 3.

*Table 3.* Default unit statistics used in TABX. Negative attack damage values correspond to healing effects.

| Name | Texture | Health | Body Radius | Body Weight | Speed | Attack Damage | Attack Range | Attack Cooldown | Space Occupied |
|---|---|---|---|---|---|---|---|---|---|
| Farmer (F) | | 60 | 1.0 | 1.0 | 1.1 | 14 | 2.5 | 2.5 | 1 |
| Assassin (S) | | 70 | 1.0 | 1.0 | 1.4 | 22 | 2.5 | 1.5 | 1 |
| TheKing (K) | | 346 | 1.47 | 10.0 | 1.2 | 46 | 3.2 | 2.5 | 1 |
| Mammoth (M) | | 685 | 4.25 | 50.0 | 1.2 | 20 | 3.0 | 6.5 | 4 |
| Archer (A) | | 40 | 1.0 | 1.0 | 1.0 | 28 | 27.0 | 8.0 | 1 |
| Cannon (C) | | 100 | 1.0 | 5.2 | 0.5 | 80 | 40.0 | 10.0 | 1 |
| Deadeye (D) | | 40 | 1.0 | 1.0 | 1.1 | 25 | 20.0 | 8.0 | 1 |
| Healer (H) | | 25 | 1.0 | 1.0 | 1.0 | -7 | 10.0 | 2.0 | 1 |
| Paladin (P) | | 220 | 1.32 | 8.5 | 1.2 | -6 | 7.5 | 2.0 | 1 |

## A.7. Zones

We introduce three distinct terrain zones, each featuring specialized functional effects designed to catalyze the diversity of emergent strategies. Each zone is defined by a triplet of components: the zone type, the elliptical geometry (specified by center coordinates $(x, y)$ and the lengths of the major and minor axes), and the magnitude of the functional effect.

- **Lava**: The lava is represented as a high-hazard region where agents incur a scalar penalty to their health attribute $H$ per time step $\Delta t$.

- **Bush**: The bush provides conditional concealment, hiding units from the observations of opposing agents unless they occupy the same zone or engage in combat.

- **Swamp**: The maximum speed of each unit is scaled by a coefficient $\mu_{\text{swamp}} \leq 1.0$ to simulate high-drag movement.

Bush zones facilitate asymmetric information flow by masking the presence of internal units from external adversaries. While teammates maintain shared observability, an agent within a bush remains hidden to opponents unless they occupy the same zone. This concealment is temporally revoked upon combat engagement—whether the unit initiates an attack or sustains damage—thereby introducing a dynamic POMDP that necessitates sophisticated strategic reasoning.

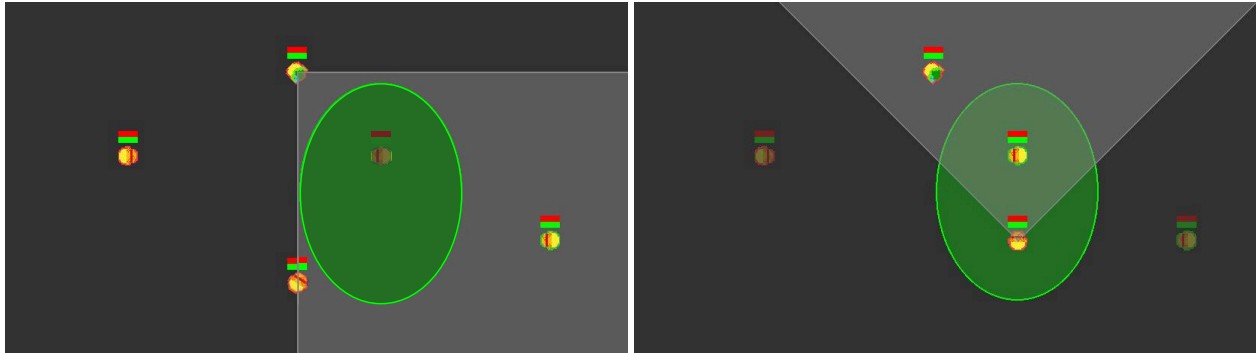

*Figure 12.* Asymmetric Visibility and the Bush Zone Effect. Allies are denoted by a red outline, while enemies are indicated by a green outline. A unit stationed within a bush is occluded from the field of view of opposing units, yet remains fully observable to its teammates (Left). This occlusion is selectively bypassed for units occupying the same bush zone, who maintain mutual observability regardless of their team affiliation (Right).

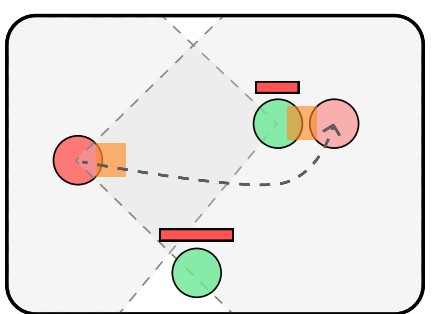 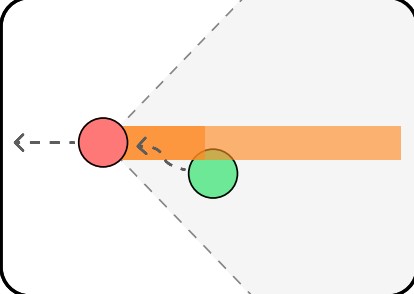 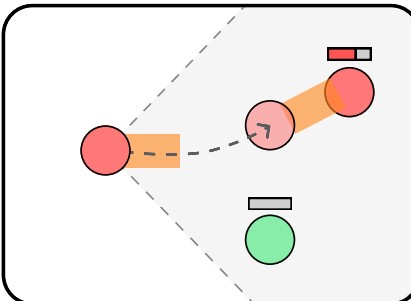

Assassin prioritize targeting the opponent with the lowest health capaticy within their field of view, attempting to remain outside the target's view field.

Ranged units attempt to maintain a separation from opponent of at least a fixed proportion of their attack range.

Healer units track damaged allies and position themselves at a distance while keeping patients within their hurtbox.

*Figure 13.* Operation of the TABX heuristic policy. Units of the same color belong to the same team. The gray region indicates a unit's field of view, and the orange region denotes its attack range.

## A.8. Role-Appropriate Heuristic Policy Mechanism

To leverage unit-specific statistical advantages, we define three orthogonal role classes: Ranger, Assassin, and Healer. These roles are mapped from intrinsic unit attributes, namely attack range, movement speed, and attack damage polarity. These classes are non-mutually exclusive, allowing a single agent to embody multiple roles—for instance, a high-mobility unit with long-range capabilities is categorized as both a Ranger and an Assassin. Each role imparts a distinct behavioral tendency; the agent's final action emerges from the composition of these role-specific logics within a unified decision-making pipeline.

The three role classes modulate the decision pipeline at three hierarchical stages: target selection, spatial positioning, and threat mitigation.

- **Target Selection**: The Target Selection mechanism employs role-specific heuristics to prioritize entities within an agent's field of view (FoV). Healer prioritize the proximal injured ally; if no injured units are detected, they default to the nearest allied unit. Assassin target visible enemies by prioritizing the lowest maximum health, using proximity as a tie-breaker. All other agents follow a Nearest-Neighbor heuristic, targeting the closest visible opponent.

- **Spatial Positioning**: During the Spatial Positioning phase, agents calculate target coordinates based on their role-specific tactical objectives. Assassin prioritize posterior positioning relative to the target's orientation to exploit positional vulnerabilities. Conversely, Healer minimize the Euclidean distance to their target to maximize healing coverage, while the remaining units seek frontal engagement to maintain a direct line of sight and combat pressure, whereas other units move toward the front of the target to maintain direct engagement.

- **Threat Mitigation**: The Threat Response stage for Ranger is characterized by a distance-maintenance protocol and

strategic environmental biasing. Agents dynamically regulate their proximity to threats by executing a retreat when opponents breach a critical threshold of their effective attack range, thereby preserving a standoff advantage. In the absence of viable targets, Ranger exhibit a proactive bias toward environmental 'bush zones' to leverage occlusion and visibility advantages, optimizing their positioning for subsequent engagement cycles.

The resulting policy follows a unified priority-based hierarchy shared across all unit types. At the highest priority, an agent executes an attack if a target resides within its effective engagement region and the attack cooldown has lapsed; otherwise, it performs a rotation to align its heading with a selected visible target if rotating would bring the selected visible target into this region. If neither condition is satisfied, Ranger units implement a kiting protocol to maintain a standoff advantage under threat. If no kiting is triggered, agents execute navigation toward visible role-dependent targets. When targets are occluded, agents utilize a memory-based pursuit mechanism targeting the last known coordinates. Lacking both visible and remembered targets, agents default to a search rotation. In this case, Ranger units first navigate toward nearby bush zones to seek concealment and, upon entering a bush zone, perform the same rotational search, whereas other units immediately rely on a stochastic rotational search. To ensure limited stochasticity, we apply an $\varepsilon$-greedy override to the final action selection.

## B. GUI Scenario Editor

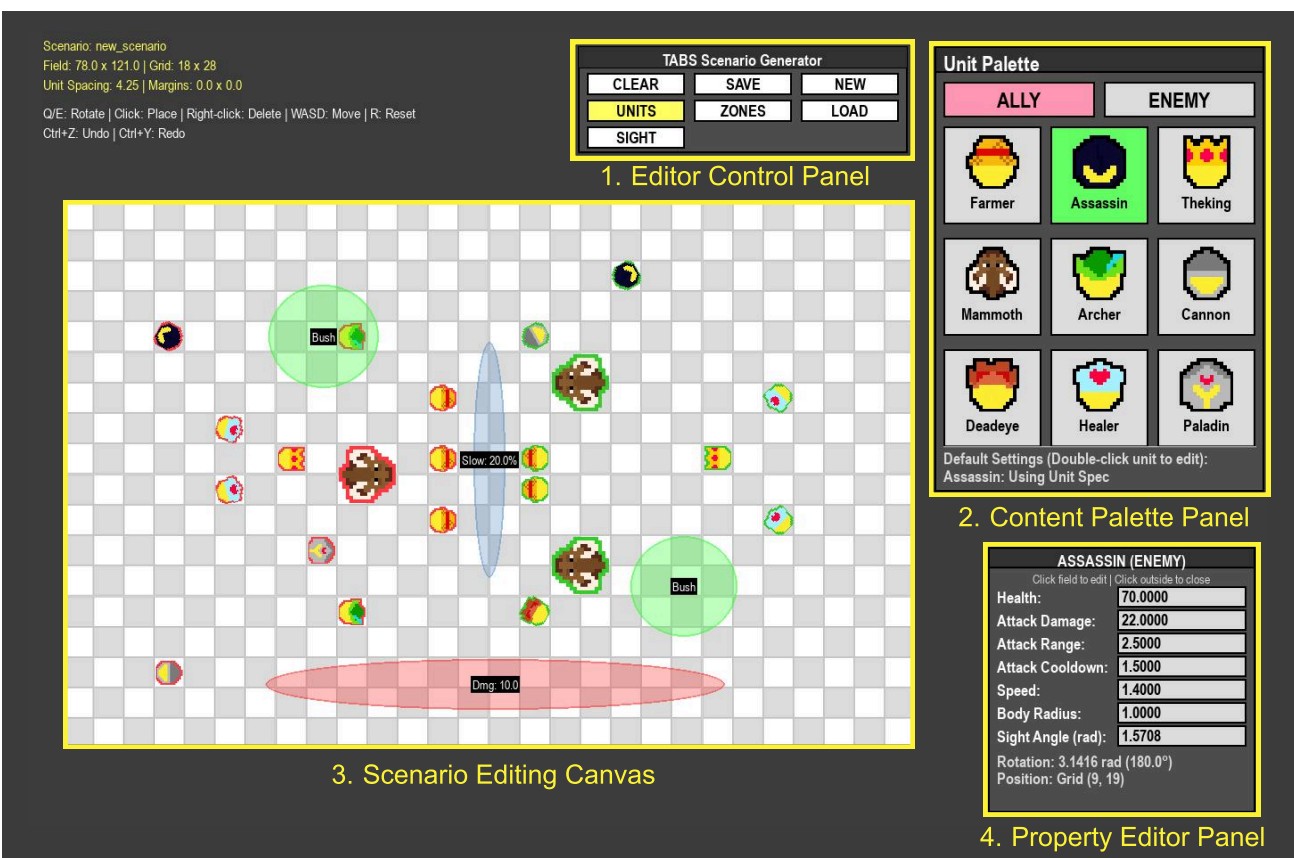

*Figure 14.* Overview of the scenario editor interface. The interface consists of (1) an editor control panel, (2) a content palette panel, (3) a scenario editing canvas, and (4) a property editor panel. Yellow bounding boxes and labels are used to highlight the main components of the editor. Allies are denoted by a red outline, while enemies are indicated by a green outline.

Figure 14 shows an overview of the GUI scenario editor used for authoring scenarios. The editor is composed of four main components: an editor control panel, a content palette panel, a scenario editing canvas, and a property editor panel. This section describes the functionality of each component and explains how they are used to construct and configure scenarios.

## B.1. Editor Control Panel

The editor control panel provides high-level operations for managing scenarios and configuring the editor state. It includes functions for creating, loading, saving, and clearing scenarios, as well as controls for switching editing modes and adjusting global visualization settings.

The *New* button opens a dialog for initializing a new scenario. In this dialog, users can specify the spatial configuration of the scenario, including the maximum field width and height, margin sizes, and unit spacing. These parameters define the underlying grid layout and determine how elements are arranged within the scenario. Once confirmed, the editor initializes an empty scenario based on the specified settings. The *Load* button allows users to load previously saved scenarios. Scenarios are organized by category (e.g., challenges, units, and zones) and can be selected through a searchable list. Loading a scenario replaces the current editor state with the stored configuration. The *Save* button opens a dialog for storing the current scenario. Users can specify the scenario name and select a target folder corresponding to different scenario types. The editor supports saving full scenarios that

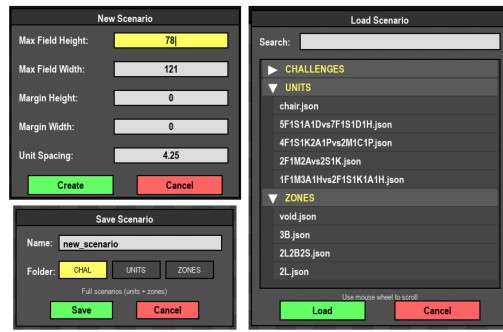

*Figure 15.* Dialog windows of the editor control panel for scenario management. The figure shows the interfaces for creating a new scenario, loading an existing scenario, and saving the current scenario, along with their corresponding configuration options.

include both unit and zone configurations. The *Clear* button removes all elements from the current scenario, resetting the editor to an empty state while preserving the global layout parameters. In addition to scenario management, the editor control panel provides mode-switching functions. The *Units* and *Zones* buttons change the active content palette, allowing users to select and place different types of editable elements. The *Sight* button adjusts the global visibility settings by toggling the viewing ranges of all units, enabling users to inspect perception-related properties during scenario design.

## B.2. Content Palette Panel

The content palette panel provides access to the editable elements that can be placed within a scenario. Depending on the current editing mode, the panel presents either unit elements or environmental zone elements, enabling users to select and configure components prior to placement.

When the *Units* mode is active, the content palette displays available unit types categorized by allegiance (i.e., ally and enemy). Users can select a unit from the palette and place it onto the scenario editing canvas by clicking on the desired location. Double-clicking a unit entry opens a configuration interface that allows users to modify the unit's attributes, such as combat-related statistics, before placement. This enables fine-grained control over unit behavior and properties during scenario design.

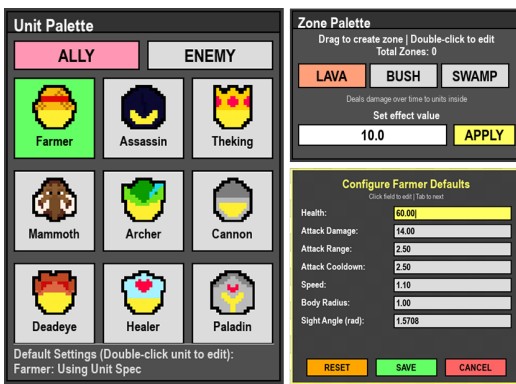

*Figure 16.* Unit and zone configuration interfaces in the content palette panel. The unit palette enables the selection of ally and enemy units and supports attribute customization through a configuration dialog, whereas the zone palette allows users to define environmental zones with adjustable effect values.

When the *Zones* mode is selected, the content palette presents environmental zone types, including lava, bush, and swamp. Zone elements are placed onto the editing canvas using a drag-and-drop interaction, allowing users to define spatial regions directly. For each zone type, users can adjust effect-related parameters, such as effect strength or influence values, which determine how the zone affects units within its area. These configurable effect values support the creation of diverse environmental conditions in scenarios.

## B.3. Scenario Editing Canvas

The scenario editing canvas serves as the primary workspace for placing and manipulating elements within a scenario. Users can directly interact with units and zones on the canvas to inspect and adjust their properties.

Double-clicking an element on the canvas opens the property editor panel, allowing users to modify the corresponding

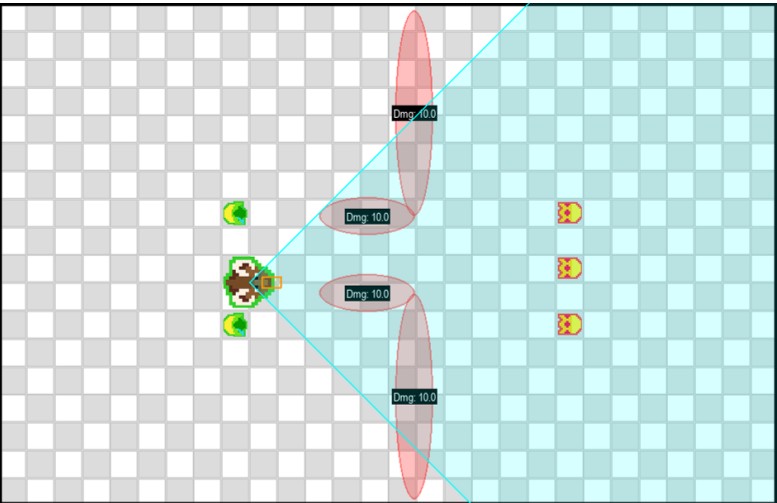

*Figure 17.* Example of interactions on the scenario editing canvas. The figure illustrates unit placement, visualization of unit fields of view, and orientation adjustment for inspecting perception- and direction-related properties.

attributes and configuration parameters. In addition, hovering the mouse cursor over a unit visualizes its field of view, enabling users to inspect perception-related properties in the scenario.

The orientation of units can be adjusted using keyboard controls. Specifically, users can rotate selected units by pressing the *Q* and *E* keys, which facilitates precise control over directional attributes during scenario design.

### B.4. Property Editor Panel

The property editor panel provides a context-sensitive interface for inspecting and modifying the attributes of selected elements. The contents of the panel are dynamically updated based on the type of element selected on the scenario editing canvas.

When a unit is selected, the panel displays unit-specific attributes, including health, attack-related parameters, movement speed, physical size, and perception properties. These parameters can be directly edited to fine-tune unit behavior and characteristics within the scenario. Additional information, such as the unit's position and orientation, is displayed to support precise spatial configuration.

When a zone element is selected, the property editor panel presents zone-specific properties, such as spatial dimensions, position, and effect-related parameters. Users can adjust effect values to control how the zone influences units within its area. This unified editing interface enables consistent manipulation of both unit and environmental properties during scenario design.

## C. Scenarios

In this section, we present several predefined scenarios designed for researchers. These scenarios are constructed specifically for the proposed research framework. We categorize the scenarios into three types: *challenges*, *unit scenarios*, and *zone scenarios*.

*Challenges* are scenarios that include fixed specifications for both units and zones. As a result, they are particularly suitable for task-specific evaluations (e.g., exploration or centralized value evaluation), where neither the units nor the zones are intended to be replaced.

In contrast, *unit scenarios* and *zone scenarios* are designed to be modular and interchangeable. Since units and zones can be independently replaced, these scenarios are well suited for evaluating generalization across different configurations. In particular, unit scenarios can be attached to arbitrary zone scenarios, enabling systematic generalization experiments.

We follow a consistent naming convention for these scenarios. *Challenges* are named without underscores, whereas *unit scenarios* and *zone scenarios* follow the format `{unit_scenarios}_{zone_scenarios}` to explicitly reflect their compositional structure. All scenario snapshots are provided in Figure 18.

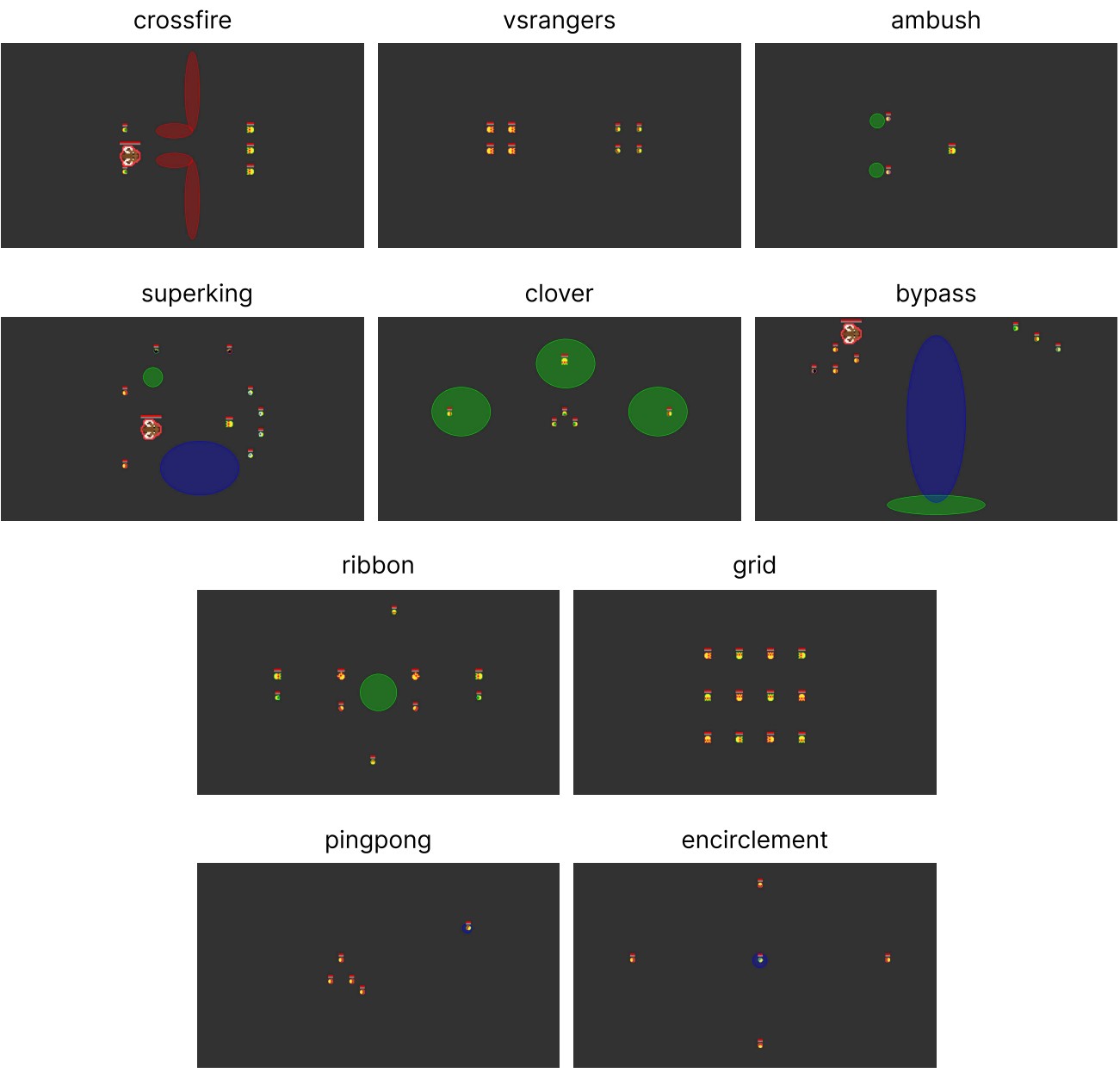

*Figure 18.* Initial state configurations across all scenarios. Snapshots illustrate the starting state distributions of agents and the diverse placement of environmental primitives.

## C.1. Challenges

`crossfire`. This scenario is designed to evaluate agent behavior in environments where terrain hazards play a central strategic role. Agents are required to actively exploit lava zones to gain positional advantages while avoiding sustained damage. Importantly, the spatial configuration allows each agent to infer most of the adversarial state through its own local observation, resulting in minimal reliance on global state information. As such, this scenario serves as a controlled setting where centralized value learning is expected to provide little to no advantage over decentralized approaches.

`vsrangers`. This scenario examines whether agents can learn effective engagement strategies against long-range opponents using short-range units. At lower difficulty levels, enemy ranged units exhibit limited kiting behavior, making the task relatively easy to learn. As the difficulty increases, however, enemies actively maintain distance, significantly increasing the challenge of initiating successful engagements. Similar to `crossfire`, accurate decision-making can largely be achieved

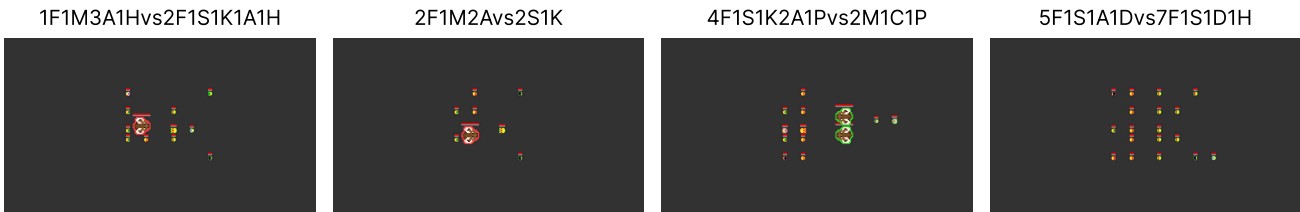

*Figure 19.* Representative initial configurations of *unit scenarios*. Snapshots illustrate diverse agent compositions under fixed unit specifications, while environmental zone layouts remain unspecified and interchangeable.

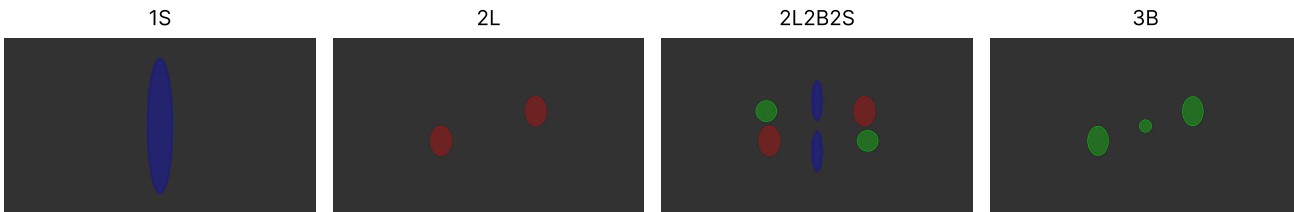

*Figure 20.* Representative initial configurations of *zone scenarios*. Snapshots illustrate diverse terrain layouts and environmental primitives, while agent compositions are left unspecified and can be freely combined with different unit scenarios.

from local observations alone, rendering global state information weakly informative.

`ambush`. This scenario focuses on asymmetric engagements under partial observability. Two cannon units must leverage bush zones to confront an enhanced king unit, whose initial maximum health is set to 800. The use of concealment is critical for success, as direct confrontation is highly unfavorable. Despite the increased difficulty, the tactical requirements remain locally grounded, and access to the global state provides limited additional benefit for value estimation.

`superking`. This scenario is designed to evaluate strategic coordination and target prioritization under support dynamics. Agents must engage an enemy king that is actively supported by healer units, making direct attacks on the king ineffective unless the supporting healers are eliminated first. Successful policies therefore require coordinated focus-fire strategies and an understanding of inter-unit dependencies, rather than reliance on environmental complexity.

`clover`. In this scenario, agents begin the episode positioned back-to-back with divergent orientations, resulting in severely disjoint initial fields of view. As a consequence, critical information about enemy positions and team-wide threat levels is not locally observable by individual agents. This scenario explicitly amplifies the importance of global state information, making it a favorable setting for centralized value learning and coordination-aware critics.

`bypass`. This scenario emphasizes strategic navigation and terrain-aware decision-making. Agents are required to exploit swamp and bush zones to avoid unfavorable direct engagements and to maneuver around adversaries. Success depends on planning paths that leverage terrain-induced constraints and visibility asymmetries, highlighting the interaction between spatial reasoning and tactical combat behavior.

`ribbon`. This scenario shares structural similarities with `clover` in that agents are initialized with disjoint fields of view, limiting the availability of local information. However, unlike `clover`, agents must additionally account for enemies that can approach from oblique angles outside their immediate view. As a result, agents are required to remain vigilant to threats emerging from blind spots, increasing the importance of global state information for accurate value estimation and coordinated responses.

`grid`. In this scenario, multiple king units are arranged in an interwoven grid-like formation, creating dense spatial entanglement among high-value targets. Effective performance requires agents to coordinate their attacks and agree on which king unit to focus on, as uncoordinated damage is insufficient to eliminate any single target efficiently. This scenario therefore emphasizes cooperative target selection and synchronization among agents.

`pingpong`. This scenario is designed as a long-horizon exploration task with sparse rewards. A stationary ranged enemy unit is placed such that agents must carefully approach and land a precise hit to obtain any positive reward. However, entering the enemy's attack range incurs sustained negative rewards, forcing agents to traverse a region of unfavorable returns before discovering a rewarding strategy. The fixed position of the enemy unit restricts possible approaches, making

naive exploration particularly ineffective.

`encirclement`. Similar to `pingpong`, this scenario features a stationary ranged enemy unit and sparse reward signals, requiring agents to engage in long-horizon exploration. However, in contrast to `pingpong`, agents can approach the enemy from multiple angles, including blind spots outside the enemy's primary attack direction. This additional flexibility reduces the effective difficulty of exploration, making the scenario moderately easier than `pingpong` despite sharing a similar reward structure.

### C.2. Unit and Zone Scenarios

In contrast to challenge scenarios with fixed configurations, unit and zone scenarios are designed to support systematic generalization studies. Unit scenarios specify agent compositions while leaving environmental layouts unspecified, whereas zone scenarios define terrain configurations independently of unit specifications. By decoupling agent embodiment from environmental structure, these scenarios enable controlled evaluation of how learned policies transfer across unseen combinations of units and zones.

Figures 19 and 20 illustrate representative initial configurations of unit and zone scenarios, respectively. Each scenario can be composed by pairing a unit scenario with an arbitrary zone scenario, allowing a combinatorial construction of evaluation environments without additional environment design effort.

## D. Comparison on Matched Scenarios

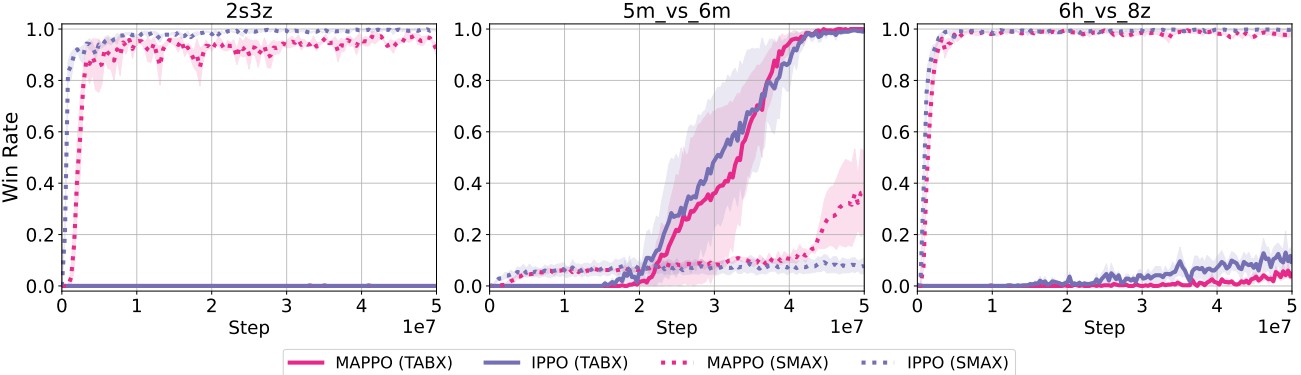

*Figure 21.* Learning curves of MAPPO and IPPO on matched scenarios in TABX and SMAX (`2s3z`, `5m_vs_6m`, `6h_vs_8z`).

We conduct direct comparisons of MAPPO and IPPO between TABX and SMAX on matched scenarios, where we aligned unit specifications and action dynamics to match SMAX as closely as possible. The enemy heuristic policy was set to medium, as in our main experiments. TABX proves harder to solve than SMAX in most scenarios, which we attribute to its more complex low-level dynamics—agents must learn precise positioning and orientation rather than simply closing the distance as in SMAX. In `2s3z`, melee units face additional difficulty as the non-targeting attack mechanism requires them to physically close in and land hits to receive any learning signal, and in `6h_vs_8z`, agents must simultaneously manage distance, rotation, and attack timing. The exception is `5m_vs_6m`, where TABX initially underperforms SMAX due to the difficulty of learning these dynamics, but surpasses SMAX after approximately 2M steps. These results demonstrate that TABX induces qualitatively distinct learning dynamics from SMAX, rather than simply adding implementation complexity.

## E. Experimental Details

In TABX, for multi-agent reinforcement learning (MARL), we provide Independent PPO (IPPO) (De Witt et al., 2020), MAPPO (Yu et al., 2022), Independent Q-Learning (IQL) (Tampuu et al., 2017), and QMIX (Rashid et al., 2020). For unsupervised environment design (UED), we provide Domain Randomization (DR) (Jakobi, 1997; Sadeghi & Levine, 2016), PLR (Jiang et al., 2021a), Robust PLR (PLR$^{\perp}$) (Jiang et al., 2021b), ACCEL (Parker-Holder et al., 2022), and SFL (Rutherford et al., 2024a), all of which are integrated with the MAPPO framework.

We train all algorithms using 128 parallel environments. For MARL experiments, agents are trained for $5 \times 10^7$ steps on

`crossfire`, `vsrangers`, `ambush`, and `superking`, and $10^8$ steps on `clover`, `bypass`, `ribbon`, and `grid`. For UED experiments, agents are trained for $5 \times 10^7$ steps. For the `ribbon` and `grid` scenarios, training is extended to a total of $3 \times 10^7$ environment steps. Experiments are conducted on a single NVIDIA RTX 4090 GPU, an AMD Ryzen Threadripper PRO 7975WX CPU (32 cores) and 512GB of RAM. Our software stack includes Python 3.10, with core implementations utilizing JAX (v0.6.1+), Flax (v0.10.6+), and Optax (v0.2.5+).

In our UED framework, we represent each level $\theta$ as the environment parameters that is passed to the environment during the reset procedure. Each level $\theta$ is sampled from a composite configuration space $\Theta$, comprising unit specifications, spatial zone parameters, heuristic policy coefficients, and physical environment constants. Specifically, TABX exposes these levels through two independent categories of free parameters: unit specifications, zone layouts, and heuristic policy parameters. Under unit specifications, we parameterize intrinsic attributes including health points, movement speeds, and attack damages. For terrain zones, the configurable parameters define the environmental factors, including zone types, spatial coordinates, axial dimensions, and associated effect magnitudes. For heuristic policy configurations, the free parameters include the stochasticity of their action selection and aggressive threshold for kitting behavior of Ranger. For the ACCEL implementation, we introduce three distinct mutation operators $\mathcal{M}$ designed to stochastically perturb the environment configuration:

- **Parameter Perturbation**: The addition of uniform noise to the continuous free parameters across all categories.

- **Spatial Transformation**: The swapping of axial dimensions within zone layouts to alter environmental factors.

- **Categorical Reassignment**: The stochastic re-indexing of zone types to modify local environment effects.

1F1M3A1Hvs2F1S1K1A1H_2L2B2S

2F1M2Avs2S1K_2L2B2S

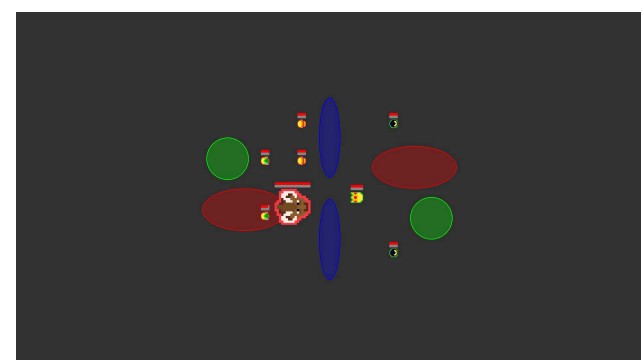

4F1S1K2A1Pvs2M1C1P_2L2B2S

5F1S1A1Dvs7F1S1D1H_2L2B2S

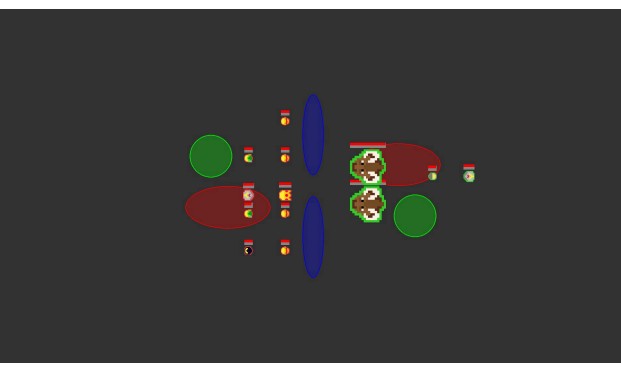

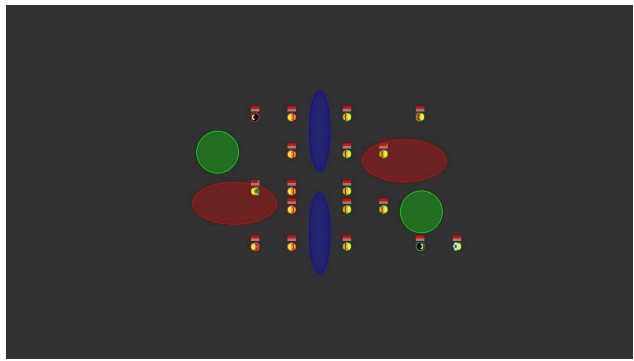

*Figure 22.* Snapshots of the four distinct UED training configurations. Scenarios are labeled according to their agent nomenclature, denoting the specific ally and enemy compositions alongside their respective environmental zone configurations.

We construct four distinct training configurations by pairing heterogeneous unit compositions with six environmental primitives, comprising two lava, two bush, and two swamp zones. These configurations are designed to provide a diverse initial distribution for the curriculum. Representative snapshots of these training scenarios are illustrated in Figure 22.

We evaluate all UED baselines on a held-out suite of unseen scenarios, which serve as controlled variants of the training configurations. To assess zero-shot generalization, we generate evaluation scenarios by maintaining invariant ally counts and environmental zone configurations while holding enemy compositions constant across the test suite.

We adopt a compositional notation to uniquely identify each scenario based on its agent and environmental constituents. For instance, the scenario `2F1M2Avs2S1K_2L2B2S` denotes an ally composition of two Farmers (F), one Mammoth (M), and two Archers (A), pitted against an enemy team of two Assassins (S) and one King (K). The environmental context is defined by the zone configuration suffix, indicating two Lava (L), two Bush (B), and two Swamp (S) zones. A comprehensive list of all evaluation scenarios and snapshots is provided in Table 4, Figure 23, and Figure 24.

*Table 4.* Evaluation scenarios of each training configurations in UED.

| Training scenario | Free parameter | Evaluation scenarios |
|---|---|---|
| 1F1M3A1Hvs2F1S1K1A1H_2L2B2S | Unit spec | (A) 1F1M3A1Hvs2F1S1K1A1H_2L2B2S
(B) 2F1M1A1C1Pvs2F1S1K1A1H_2L2B2S
(C) 1S1M1A2C1Hvs2F1S1K1A1H_2L2B2S
(D) 1F1K2D2Pvs2F1S1K1A1H_2L2B2S |
| | Zone | (A) 2F1S1K1A1H_2L2B2S
(B) 2F1S1K1A1H_2L2B2S-1
(C) 2F1S1K1A1H_2L2B2S-2 |
| 2F1M2Avs2S1K_2L2B2S | Unit spec | (A) 2F1M2Avs2S1K_2L2B2S
(B) 2K1M2Dvs2S1K_2L2B2S
(C) 1M4Avs2S1K_2L2B2S
(D) 1S3K1Cvs2S1K_2L2B2S |
| | Zone | (A) 2F1M2Avs2S1K_2L2B2S
(B) 2F1M2Avs2S1K_2L2B2S-1
(C) 2F1M2Avs2S1K_2L2B2S-2 |
| 4F1S1K2A1Pvs2M1C1P_2L2B2S | Unit spec | (A) 4F1S1K2A1Pvs2M1C1P_2L2B2S
(B) 2F2S1K1M2C1Pvs2M1C1P_2L2B2S
(C) 4F1S1K1C1Pvs2M1C1P_2L2B2S
(D) 3F1S1K1A1D1Pvs2M1C1P_2L2B2S |
| | Zone | (A) 4F1S1K2A1Pvs2M1C1P_2L2B2S
(B) 4F1S1K2A1Pvs2M1C1P_2L2B2S-1
(C) 4F1S1K2A1Pvs2M1C1P_2L2B2S-2 |
| 5F1S1A1Dvs7F1S1D1H_2L2B2S | Unit spec | (A) 5F1S1A1Dvs7F1S1D1H_2L2B2S
(B) 4F1S1A1Cvs7F1S1D1H_2L2B2S
(C) 2F1S1A1C1Dvs7F1S1D1H_2L2B2S
(D) 3F2S1K1A1Cvs7F1S1D1H_2L2B2S |
| | Zone | (A) 5F1S1A1Dvs7F1S1D1H_2L2B2S
(B) 5F1S1A1Dvs7F1S1D1H_2L2B2S-1
(C) 5F1S1A1Dvs7F1S1D1H_2L2B2S-2 |

## F. Baseline Implementations

Our implementations are built upon JaxMARL (Rutherford et al., 2024b), JaxUED (Coward et al., 2024), and SFL (Rutherford et al., 2024a), utilizing recurrent neural network (RNN) architectures to handle temporal dependencies.

### F.1. Unsupervised Enviornment Design

For regret-based methods such as PLR, PLR$^\perp$, and ACCEL, we provide two heuristic score functions for computing regret: Maximum Monte Carlo (MaxMC) and Positive Value Loss (PVL).

2F1M1A1C1Pvs2F1S1K1A1H      1S1M1A2C1Hvs2F1S1K1A1H      1F1K2D2Pvs2F1S1K1A1H

2K1M2Dvs2S1K      1M4Avs2S1K      1S3K1Cvs2S1K

2F2S1K1M2C1Pvs2M1C1P      4F1S1K1C1Pvs2M1C1P      3F1S1K1A1D1Pvs2M1C1P

4F1S1A1Cvs7F1S1D1H      2F1S1A1C1Dvs7F1S1D1H      3F2S1K1A1Cvs7F1S1D1H

*Figure 23.* Visualization of the unit variants used for zero-shot evaluation.

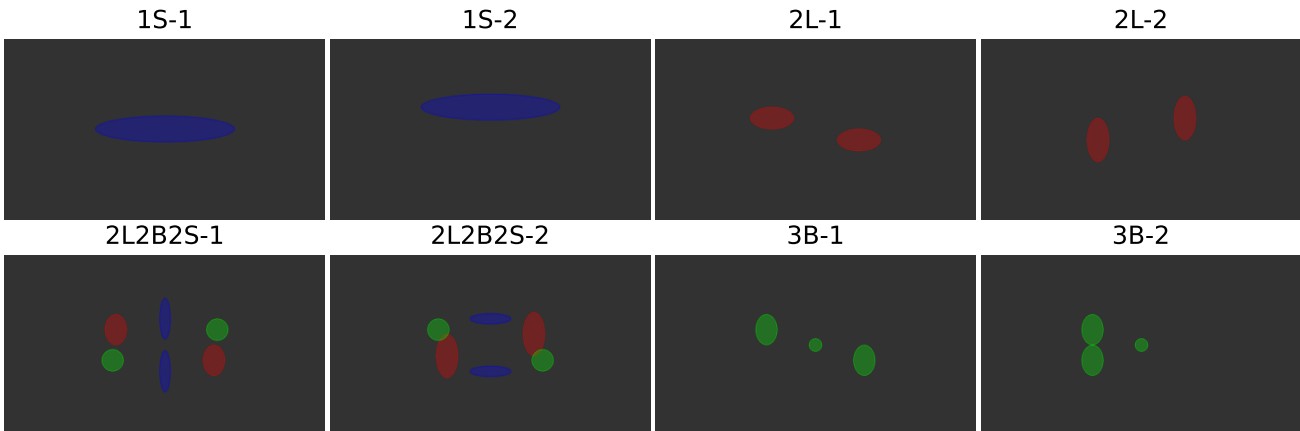

*Figure 24.* Visualization of the environmental zone configurations used for zero-shot evaluation.

- **Positive Value Loss (PVL)**: PVL serves as an approximation for regret, calculated as the mean of the positive components of the multi-step return estimates across an entire episode. Formally, we define PVL as:

$$\text{PVL} \doteq \frac{1}{T} \sum_{t=0}^{T} \max \left( \sum_{k=t}^{T} (\gamma\lambda)^{k-t} \delta_k, 0 \right), \tag{2}$$

  where $T$ denotes the episode length, $\gamma$ is the discount factor, and $\lambda$ represents the GAE parameter. The term $\delta_t \doteq R_t + \gamma V(s_{t+1}) - V(s_t)$ is the temporal difference (TD) error at time $t$, where $V(s_t)$ is the value function estimating the expected discounted return under the current policy $\pi$.

- **Maximum Monte Carlo (MaxMC)**: MaxMC utilizes the highest cumulative return achieved on a specific level as the prioritization metric: $\text{MaxMC} \doteq \frac{1}{T} \sum_{t=0}^{T} (R_{\max} - V(s_t))$

## F.2. Exploration

We incorporate Random Network Distillation (RND) (Burda et al., 2019) as a representative baseline to address the intrinsic exploration challenges inherent in our environment. Following the implementation in Matthews et al. (2024a), we integrate RND into the MAPPO framework by computing an intrinsic reward derived from the prediction error of a global state representation:

$$r_{\text{intrinsic}} = \beta \cdot \mathbb{E}_{s' \sim \tau} \|f_\phi(s) - g(s)\|^2, \tag{3}$$

where $\beta$ denotes the intrinsic reward scaling coefficient and $\tau$ represents the distribution of states within the trajectories collected by the agents. Here, $f_\phi$ is the predictor network parameterized by $\phi$, and $g$ is a fixed, randomly initialized target network.

During the MAPPO update phase, we construct the total advantage by summing the Generalized Advantage Estimates (GAE) (Schulman et al., 2015) computed independently for the extrinsic and intrinsic reward streams.

## F.3. Value Error Estimation

To analyze centralized value estimation, we compute the absolute value error $|V(s) - V^*(s)|$. We first collect 128 trajectories of length 512 using the current policy, and uniformly sample one target state from each trajectory, resulting in a set of target states $S = \{s_1, \ldots, s_{128}\}$. For each target state $s_i$, we estimate the Monte Carlo value $V^*(s_i)$ as the mean return over 128 rollouts starting from $s_i$ under the current policy. Finally, we report the mean absolute error between the learned value estimates $V(s_i)$ and the corresponding Monte Carlo estimates $V^*(s_i)$.

## F.4. Hyperparameters

*Table 5.* IPPO and MAPPO Hyperparameters

| Hyperparameter | Value |
| --- | --- |
| Optimizer | Adam (Kingma, 2014) |
| Learning rate | 4e-3 |
| Hidden dim | 128 |
| Discount factor ($\gamma$) | 0.99 |
| Batch size | 32 |
| Activation | ReLU |
| Layer norm (Ba et al., 2016) | True |
| Update epochs | 4 |
| GAE factor ($\lambda$) | 0.95 |
| Clip coefficient | 0.05 |
| Entropy coefficient ($\sigma$) | 0.01 |
| Value coefficient | 0.5 |

*Table 6.* UED Hyperparameters

| Hyperparameter | |
|---|---|
| Discount factor ($\gamma$) | 0.99 |
| GAE factor ($\lambda$) | 0.99 |
| Level buffer size | 4000 |
| Staleness coefficient | 0.3 |
| Prioritization | Rank |
| Temperature | 0.3 |
| Replay rate | 0.8 |

*Table 7.* RND Hyperparameters

| Hyperparameter | Value |
|---|---|
| Hidden dim | 128 |
| Ouput dim | 256 |
| # of layers | 3 |
| Learning rate | 3e-4 |
| GAE coefficient | 0.01 |
| Reward coefficient ($\beta$) | 0.5 |
| Entropy coefficient ($\sigma$) | 0.001 |
| Loss coefficient | 0.01 |
| Exploration update epochs | 1 |

*Table 8.* IQL and QMIX Hyperparameters

| Hyperparameter | Value |
|---|---|
| Optimizer | RAdam (Liu et al., 2019) |
| Learning rate | 5e-5 |
| Hidden dim | 512 |
| Discount factor ($\gamma$) | 0.99 |
| Batch size | 32 |
| Activation | ReLU |
| Layer norm (Ba et al., 2016) | True |
| Buffer size | 5000 |
| Reward scale | 10 |
| Target update interval | 10 |
| Update epochs | 8 |
| Mixer hidden dim | 256 |
| Mixer embedding dim | 64 |

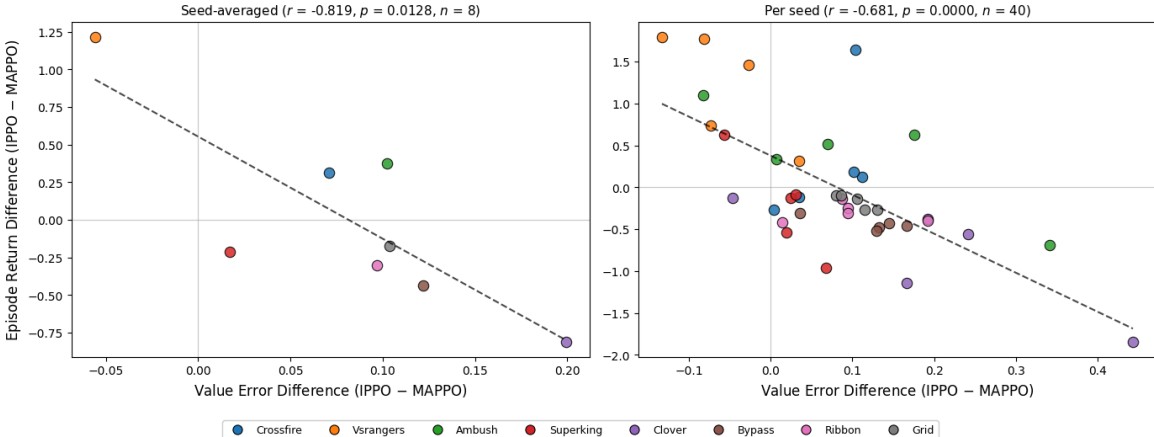

*Figure 25.* Pearson correlation between episode return difference and value error difference (IPPO − MAPPO) across all evaluation scenarios. **Left**: scenario-level correlation after averaging over seeds ($r = -0.819$, $p = 0.0128$). **Right**: seed-level correlation without averaging ($r = -0.681$, $p < 0.0001$). Both panels show a strong negative relationship, indicating that lower value error is reliably associated with higher returns.

## G. Correlation Analysis between Value Error and Episode Return

We computed the Pearson correlation between the episode return difference and value error difference (IPPO - MAPPO) across all scenarios. When aggregating over seeds, the correlation is strongly negative ($r = -0.819$, $p = 0.0128$), confirming that lower value error is reliably associated with higher returns at the scenario level. We additionally report the correlation using individual seed-level data points without averaging ($r = -0.681$, $p < 0.0001$), which corroborates the same trend, as shown in Figure 25.

## H. Time Comparison with SMAX

To isolate raw per-step throughput from JIT recompilation overhead, we report a fixed-configuration comparison in which the environment is instantiated once and stepped repeatedly without resets. Both environments are instantiated once and stepped repeatedly without scenario resets, isolating raw simulation throughput from recompilation overhead. As shown in Figure 26, TABX matches or exceeds SMAX for smaller team sizes across all parallelism levels. For larger teams, SMAX is faster at low parallelism—reflecting the higher per-step cost of TABX's richer dynamics (fan-shaped field of view, hurtbox resolution, terrain effects)—but this gap diminishes as parallelism increases. This confirms that TABX's advantage in the reconfiguration setting (Figure 10) stems from the elimination of recompilation overhead, not from inherently faster simulation.

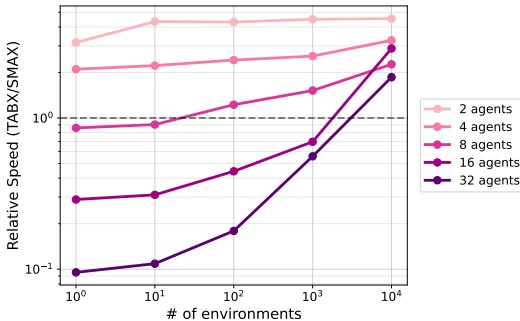

*Figure 26.* Fixed-configuration throughput comparison between TABX and SMAX.

## I. Policy vs. Policy

TABX directly supports policy-vs-policy evaluation. Since the environment accepts actions for all agents across both teams and returns per-team rewards, two independently trained policies can be assigned to opposing teams without any modification to the core environment. This enables direct algorithm comparisons such as MAPPO vs. IPPO in a competitive setting. Each policy (IPPO, IQL, MAPPO, QMIX) was independently trained against the medium-difficulty heuristic opponent for both the ally and enemy sides, then evaluated in all pairwise matchups. As a demonstration, we provide a payoff matrix from such evaluations in Figure 27.

*Figure 27.* Policy vs. policy payoff matrices (ally win rate, averaged over seeds) across all scenarios.

# J. Additional Experimental Results

## J.1. Additional Performance Metrics

We provide several interesting metrics: episode returns (Figure 28), episode lengths (Figure 29), and first-kill rate (Figure 30). In all figures, results are averaged over five random seeds, with shaded regions indicating the standard error.

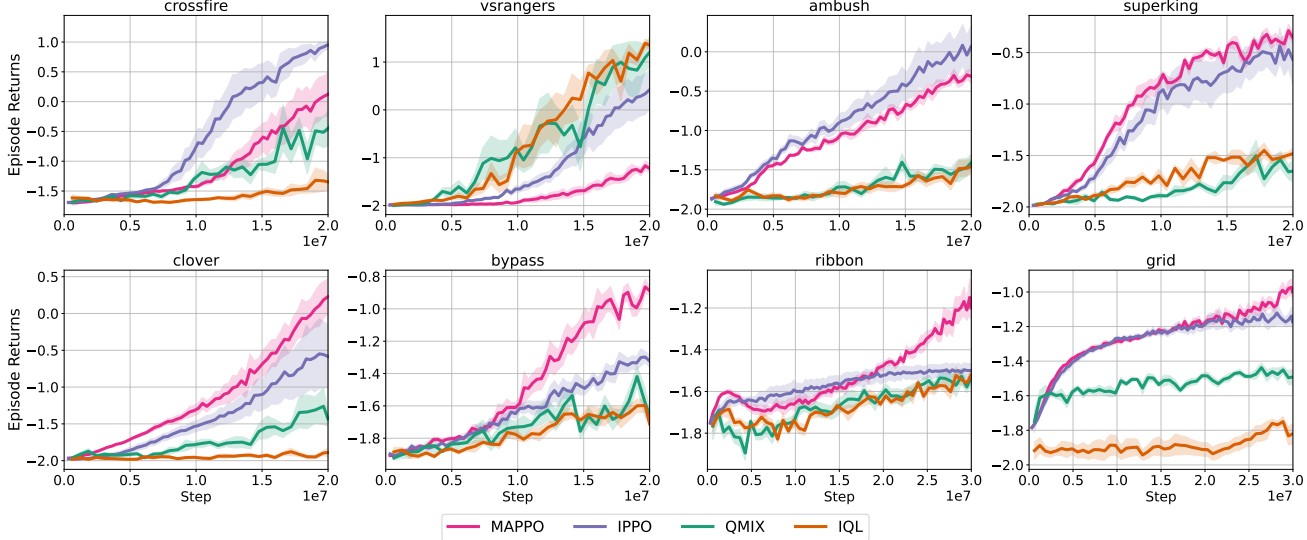

*Figure 28.* Average episode returns measured during training across evaluation scenarios.

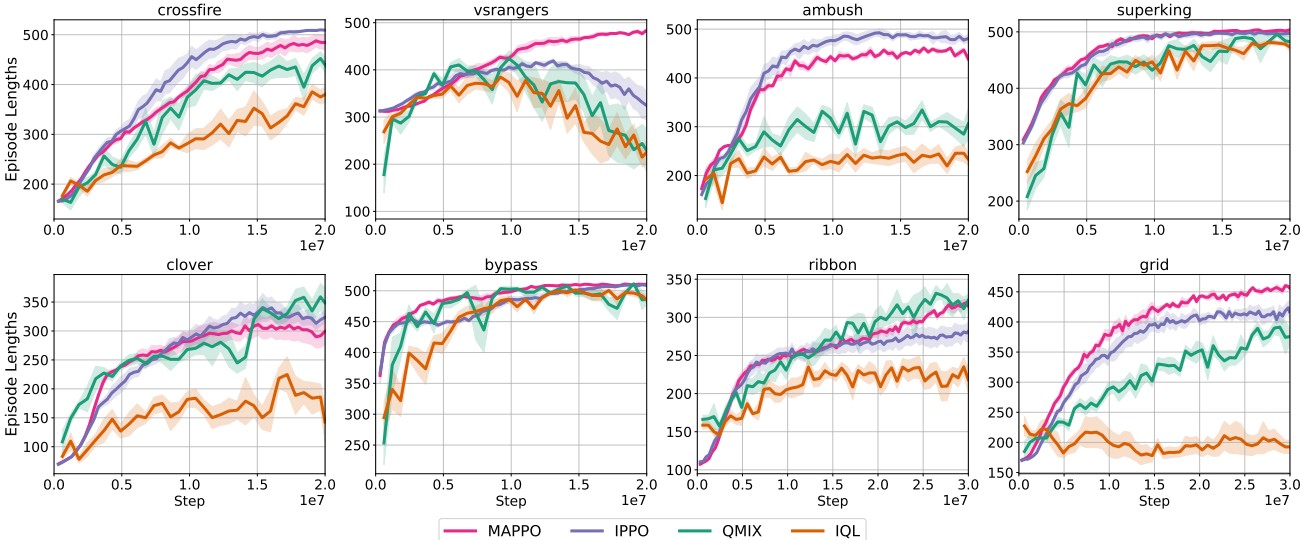

*Figure 29.* Average episode lengths measured during training across evaluation scenarios.

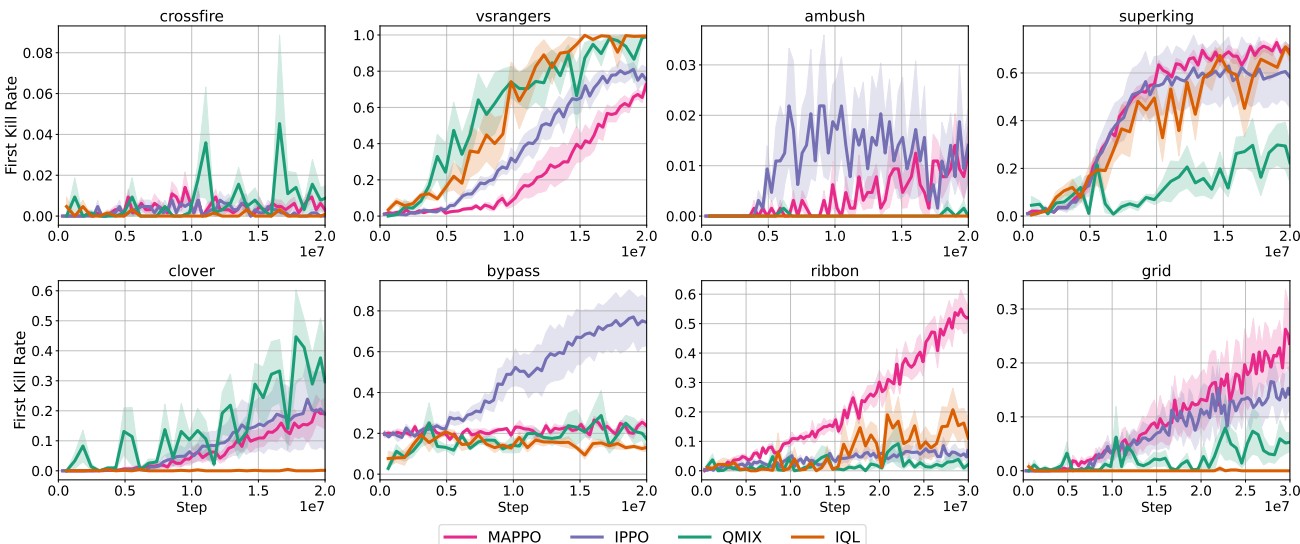

*Figure 30.* First kill rates measured during training across evaluation scenarios.

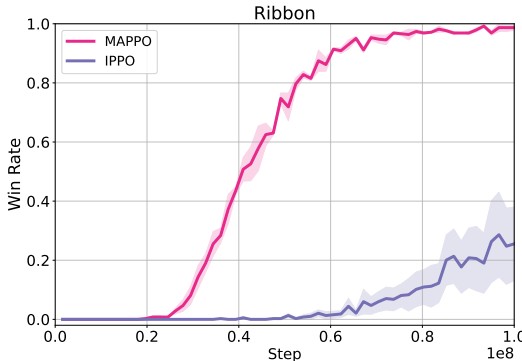

*Figure 31.* Training curves on the Ribbon environment over $10^9$ environment steps. The figure reports the same performance metrics as in the main experiments, evaluated under extended training horizons.

## J.2. Extended Training on Challenging Environments

We report additional training results on challenging scenario, `ribbon`. For these experiments, agents are trained for $10^9$ environment steps, extending the training horizon used in the main experiments in Figure 31. While MAPPO eventually achieves a win rate of nearly 100%, IPPO continues to struggle because `ribbon` scenarios are explicitly designed to require global coordination. This result highlights that TABX allows us to construct a set of scenarios that explicitly target specific MARL research questions.

## J.3. Hyperparameter Analysis of Exploration

The `pingpong` and `encirclement` scenarios are specifically designed to evaluate performance in long-horizon tasks characterized by sparse reward signals. In these environments, obtaining a positive reinforcement signal is challenging due to the high degree of precision required to successfully strike an opponent. To investigate the sensitivity of agent performance to the degree of exploration, we conduct a grid search over two key hyperparameter axes: the MAPPO entropy regularization coefficient ($\alpha$) and the RND intrinsic reward coefficient ($\beta$).

As illustrated in Figure 32, we observe a clear performance gradient: agent performance scales positively with the RND reward coefficient $\beta$ and inversely with the MAPPO entropy coefficient $\sigma$. This trend is particularly pronounced in the `encirclement` scenario, suggesting that in highly sparse reward landscapes, explicit novelty-seeking exploration (via RND) is more effective than stochastic policy noise (via entropy) for discovering the precise coordination required for task

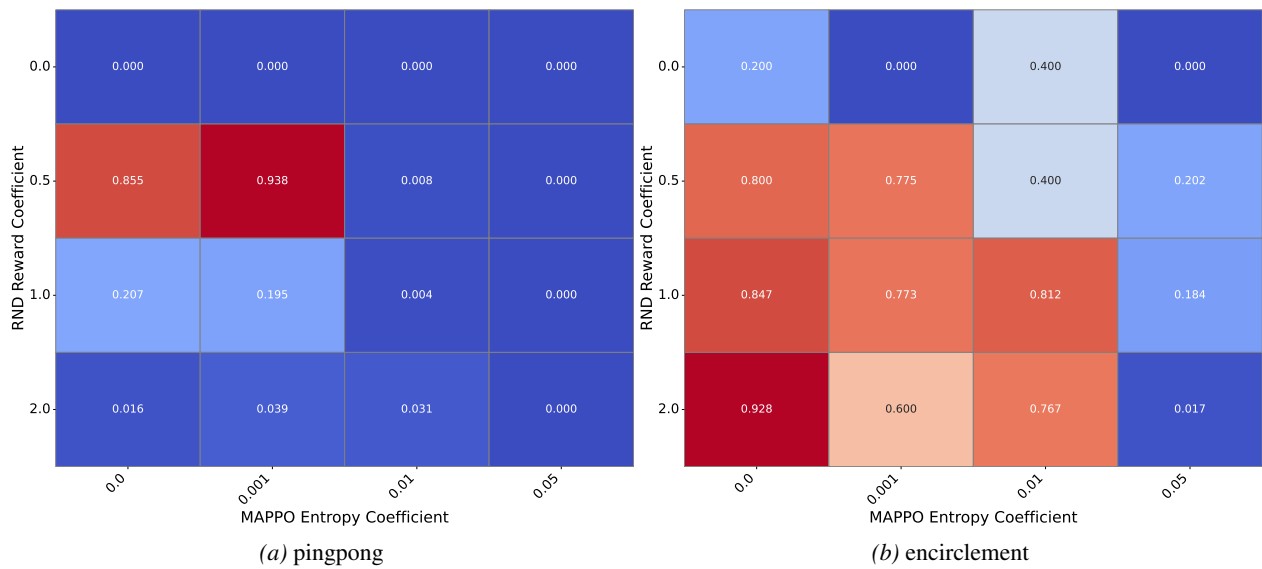

*(a)* pingpong                                                    *(b)* encirclement

*Figure 32.* Hyperparameter sensitivity analysis. Heatmap illustrating the joint effect of the RND reward scaling coefficient $\beta$ and the MAPPO entropy coefficient $\sigma$ on mean episodic return.

completion.

### J.4. Zero-shot Generalization

We report the zero-shot generalization performance of various UED algorithms—integrated within the MAPPO framework—across unit specification and environmental zone configurations in Figure 35 and Figure 36, respectively. Our results reveal an asymmetric robustness profile: while the baselines demonstrate resilience to topological perturbations (zone layouts), they exhibit significant performance degradation under unit specification shifts. These findings suggest that a comprehensive exploration of the multi-agent configuration space is critical, as narrow parameterization may fail to capture the complex dependencies inherent in heterogeneous agent interactions.

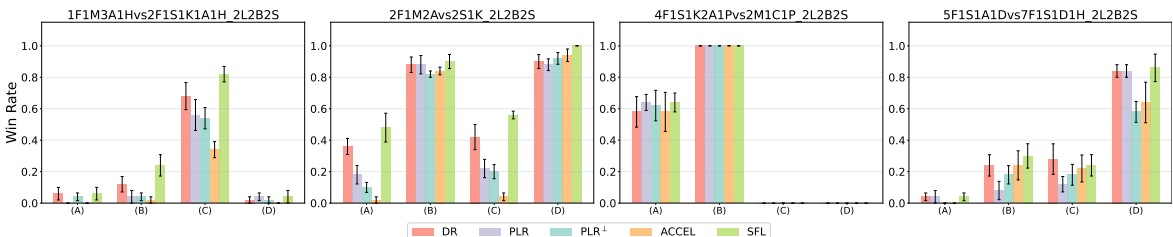

*Figure 33.* Average episodic win rates for UED baselines across all evaluation scenarios. We report the results for the four primary training configurations at the final training step, aggregating zero-shot performance across all unit specification variants.

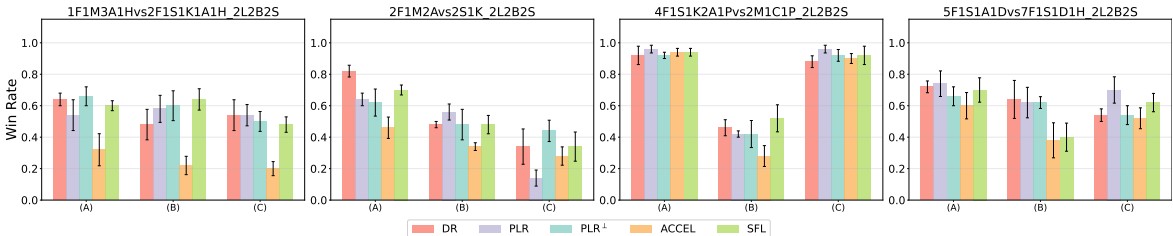

*Figure 34.* Average episodic win rates for UED baselines across all evaluation scenarios. We report the results for the four primary training configurations at the final training step, aggregating zero-shot performance across all zone configuration variants.

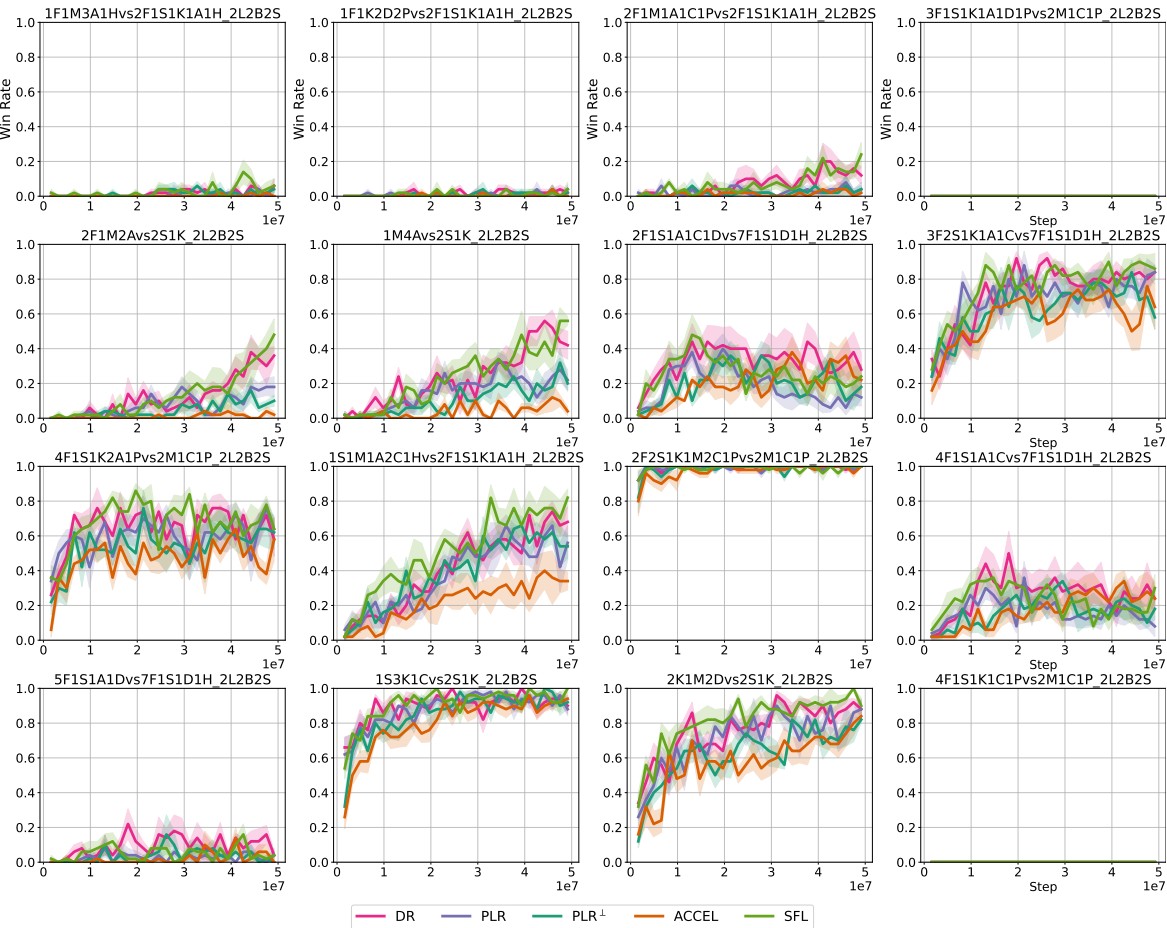

*Figure 35.* Average episodic win rates for UED baselines across all evaluation scenarios. We report the results for the four primary training configurations, aggregating zero-shot performance across all unit specification variants.

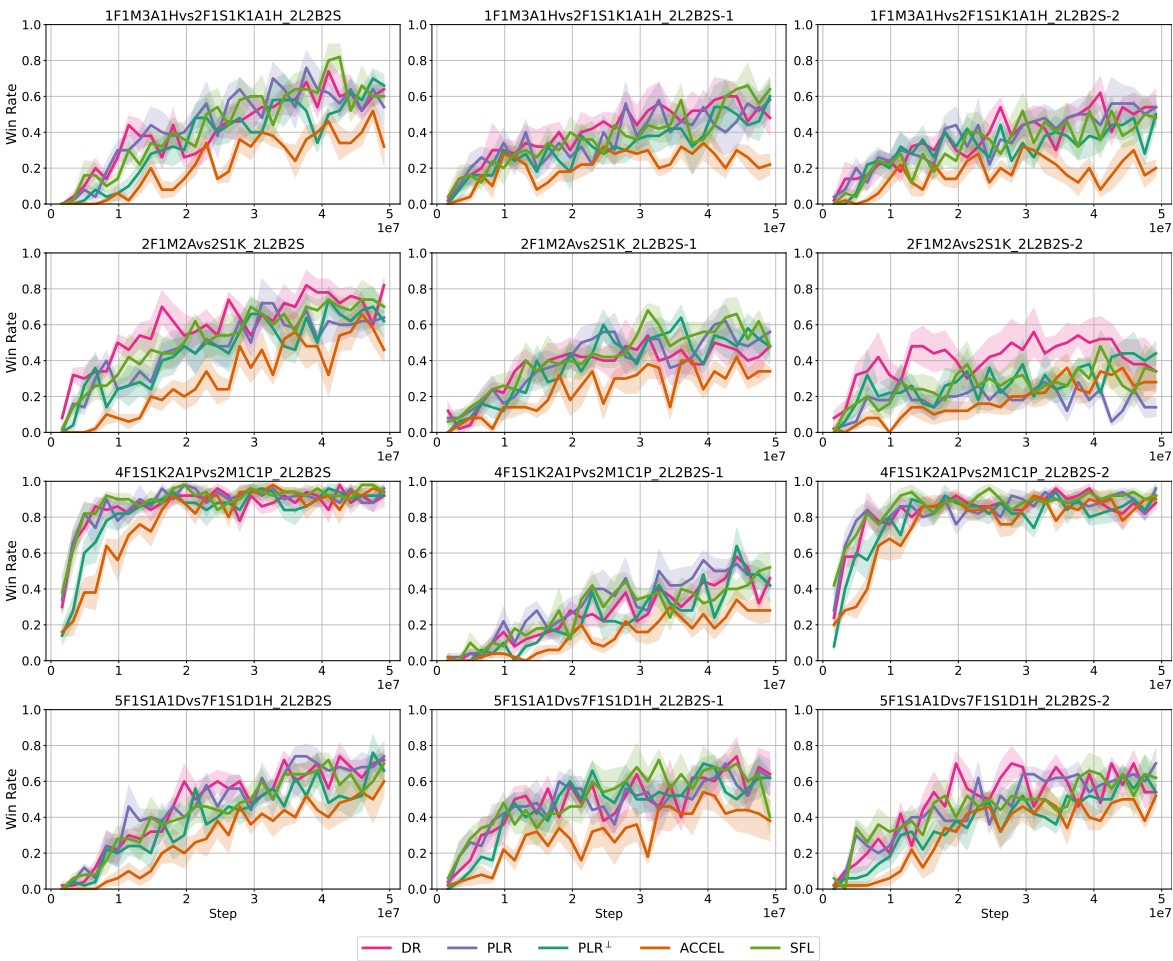

*Figure 36.* Average episodic win rates for UED baselines across all evaluation scenarios. We report the results for the four primary training configurations, aggregating zero-shot performance across all zone configuration variants.

### J.5. UED Experiments Across Heuristic Policy Difficulty Levels

We evaluate the zero-shot generalization performance of various UED algorithms across a spectrum of heuristic policy tiers, ranging from random (highest noise) to expert (lowest noise). Episode win rates are averaged across four evaluation scenarios and three seeds. Heuristic policy configuration parameters include the stochasticity of actions and an aggressiveness threshold related to "kiting" behavior. As shown in Figure 37, overall average performance is higher than in other free-parameter settings. This highlights that changes in unit attributes or environmental configurations are more critical for achieving robustness to unseen tasks.

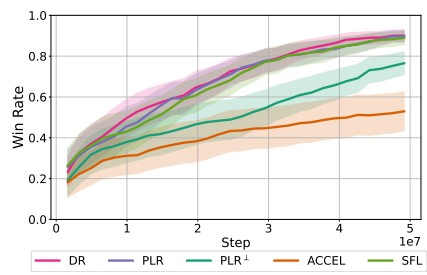

*Figure 37.* Average episode win rate across heuristic policy difficulty tiers.

### J.6. Training Wall-Clock Time

We report the end-to-end wall-clock training time of all algorithms. All measurements include environment simulation, policy evaluation, optimization, and, where applicable, level generation and replay. Experiments are conducted on a single NVIDIA RTX PRO 6000 GPU with an Intel Xeon® 6767P CPU.

For MARL experiments, agents are trained for $2 \times 10^7$ environment steps using 128 parallel environments, with the `ribbon` and `grid` scenarios trained for $3 \times 10^7$ steps. Table 9 reports wall-clock training times for MARL baselines across challenge scenarios. Policy-gradient methods (MAPPO and IPPO) require shorter training times than value-based methods (QMIX and IQL), which involve replay buffers and centralized mixing networks.

Table 10 reports training times for MAPPO with Random Network Distillation (RND) in long-horizon exploration scenarios. Table 11 summarizes wall-clock times for UED algorithms integrated with MAPPO, including replay-based methods that additionally perform level scoring and replay during training.

*Table 9.* Training Time for Challenges Scenarios

| Scenario | MAPPO | IPPO | QMIX | IQL |
|---|---|---|---|---|
| ambush | 7m | 4m | 24m | 24m |
| bypass | 9m | 5m | 32m | 32m |
| clover | 8m | 5m | 29m | 29m |
| crossfire | 8m | 5m | 29m | 29m |
| grid | 13m | 8m | 57m | 56m |
| ribbon | 12m | 8m | 51m | 49m |
| superking | 9m | 6m | 35m | 33m |
| vsrangers | 8m | 5m | 31m | 32m |

*Table 10.* Training Time for MAPPO+RND

| Scenario | Runtime |
|---|---|
| encirclement | 15m |
| pingpong | 16m |

*Table 11.* Training Time for UED Algorithms

| Scenario | Param | DR | PLR | PLR$^{\perp}$ | Accel | SFL |
|---|---|---|---|---|---|---|
| 1F1M3A1Hvs2F1S1K1A1H_2L2B2S | zone | 28m | 30m | 28m | 24m | 33m |
| | unit | 28m | 32m | 28m | 24m | 33m |
| 2F1M2Avs2S1K_2L2B2S | zone | 24m | 25m | 22m | 18m | 26m |
| | unit | 23m | 25m | 22m | 18m | 27m |
| 4F1S1K2A1Pvs2M1C1P_2L2B2S | zone | 31m | 33m | 31m | 25m | 37m |
| | unit | 30m | 33m | 30m | 26m | 37m |
| 5F1S1A1Dvs7F1S1D1H_2L2B2S | zone | 37m | 42m | 40m | 35m | 47m |
| | unit | 38m | 41m | 39m | 36m | 47m |

