# OpenReview forum: "TABX: A High-Throughput Sandbox Battle Simulator for Multi-Agent Reinforcement Learning"
_ICML.cc/2026/Conference — ICML 2026 regular_

### Official Review · Reviewer_TpjV · 2026-03-05

**Soundness:** 3
**Presentation:** 4
**Significance:** 3
**Originality:** 2
**Overall Recommendation:** 5
**Confidence:** 3

**Summary:**

The paper introduces the Totally Accelerated Battle Simulator in JAX (TABX), a high-throughput, hardware-accelerated multi-agent reinforcement learning (MARL) environment. The authors identify that existing MARL benchmarks rely on static task designs, which limits the systematic evaluation of algorithms across diverse conditions. TABX addresses this by providing a highly modular sandbox where researchers can dynamically reconfigure unit attributes, terrain zones, heuristic policies, and physical dynamics without modifying the underlying code or incurring JIT recompilation overhead. The simulator features fan-shaped partial observability and non-targeted interaction mechanics. The authors validate the environment's utility through comprehensive experiments covering centralized value learning, sparse-reward exploration, and zero-shot generalization using Unsupervised Environment Design (UED).

**Compliance With Llm Reviewing Policy:**

Affirmed.

**Final Justification:**

Authors addressed all my concerns and demonstrated the value of their work.

**Key Questions For Authors:**

- The action space is strictly discrete. Are there any plans to support continuous action spaces to broaden the applicability of TABX to a wider range of control tasks?

- Could you elaborate on the rationale behind the design of the specific scenarios presented in the paper? For instance, how were the specific unit compositions and zone placements chosen to isolate the specific MARL challenges evaluated?

**Limitations:**

- Authors evaluate the benefit in terms of time complexity of TABX with respect to SMAX. However, the action space of TABX is significantly smaller, since it adopts action masking for movements and automate the target selection to the closer enemy/allied in the field of view, while SMAC requires to select a specific target ID. I believe that the comparison is slightly unfair, as SMAC's larger action space allows for the discovery of more granular strategies at the cost of higher computational time.

- Connected to the previous point, the adoption of heuristic auto-targeting hinders the development of novel and unexpected strategies. For example, an agent cannot intentionally shoot past a frontline tank to focus-fire a high-value healer. This limitation could bottleneck research into algorithms that aim for autonomous role specialization (e.g., LOLA), as agents are prevented from executing behaviors independent of their crafted targeting heuristic.

**Strengths And Weaknesses:**

### Soundness
The paper is technically sound and robust. The methodology is supported by an extensive experimental campaign that benchmarks multiple established algorithms (MAPPO, IPPO, QMIX, IQL) and UED baselines across a systematically designed suite of scenarios.
Zero-shot generalization results could have been investigated a bit more to strengthen the analysis with further insights.

### Presentation
The paper is exceptionally well-presented and structured. The authors provide a neat, logical analysis supported by comprehensive visualizations, parameter tables, and training time reports. The supplementary materials and repository descriptions indicate a clean, well-structured codebase.
Some environment figures are difficult to read. For example, in Figure 3, it is very difficult to distinguish allied units from enemies because the colored outlines (red vs. green) are thin and not visually distinct against the dark background. Moreover, there is a typo in the legend of Fig.10 ("TABS" should be "TABX").

### Significance
The authors provide a scalable, fast, and highly customizable environment on which to train and evaluate MARL algorithms. Avoiding JIT-recompilation overhead during environment resets is a highly relevant contribution that will allow researchers to easily and efficiently adopt UED and MARL solutions at scale.

### Originality
While battleground simulators currently exist (e.g., SMAC), TABX’s most original contribution is its synthesis of a JAX-based architecture with a high degree of accessible configurability (including a GUI). This allows for faster and more scalable experiments.

---

> ### Author Rebuttal · Authors · 2026-03-31
>
> We sincerely appreciate the reviewer’s insightful comments.
>
> **Q1. Support continuous action spaces.**
>
> We plan to introduce support for hybrid action spaces that combine discrete actions (such as attacking or healing) with continuous control for movement and rotation.
>
> **Q2. The design details of TABX scenarios.**
>
> The scenarios were designed by first identifying a target MARL challenge and then constructing unit compositions and zone placements to isolate that specific variable. For instance, in the _clover_ scenario, enemies remain occluded or positioned beyond the agents' immediate perceptual fields during the initial timesteps. To demonstrate the necessity of global information, we placed bushes that hide enemies to enhance partial observability and composed the ally team of ranged units to allow them to scout the bushes more easily. We will add the scenario design details to the revised manuscript.
>
> **L1. Time complexity comparison with SMAX.**
>
> The throughput comparison in Fig.10 is conducted in the UED setting, where the dominant cost in SMAX comes from repeated JAX recompilation rather than the size of the action space. In fact, TABX's richer per-step dynamics (fan-shaped field of view, hurtbox resolution, terrain effects) are more computationally expensive than SMAX's distance-based checks, and TABX can be slower than SMAX in fixed-configuration settings with fewer parallel environments and larger team sizes, as shown in our additional comparison (https://imgur.com/a/D2hRqQs). This confirms that the comparison is not biased by action space differences.
>
> **L2. Strategic expressiveness of the attack mechanism**
>
> While our non-targeting attack/heal mechanism removes explicit target selection, it does not eliminate strategic behavior; rather, it shifts the strategic challenge toward spatial coordination and positioning. Because targets are determined by proximity and field of view, agents must actively control their positioning to influence which interactions occur. For example, rather than explicitly selecting a rear healer as a target, agents can learn to flank around front-line units to bring the healer into their attack range — a behavior that requires multi-agent coordination to execute successfully. In this sense, strategy is not reduced but restructured, emphasizing emergent spatial tactics such as front-line blocking, flanking, and focus-firing through positioning over explicit target ID selection.
>
> **W. Improving figure clarity**
>
> We will improve the clarity of the scenario snapshots by increasing the visual contrast between allied and enemy units. We also corrected the typo in Figure 10. Thank you for catching these issues.

---

> > ### Author Rebuttal · Reviewer_TpjV · 2026-04-04
> >
> > Thank you for answering my questions. I appreciate and understand your responses, therefore I will suggest acceptance.

---

### Official Review · Reviewer_uRhd · 2026-03-12

**Soundness:** 3
**Presentation:** 3
**Significance:** 3
**Originality:** 2
**Overall Recommendation:** 5
**Confidence:** 4

**Summary:**

The paper introduces TABX, a JAX-based and accelerator-compatible multi-agent battle simulator with a core focus on configurability. Its main claim is that unit specifications, terrain zones, heuristic opponent policies, and physics dynamics can be varied across episodes without code modification or JIT recompilation. The environment features fan-shaped partial observability, non-targeted hurtbox-based attacks, nine heterogeneous unit types, three terrain zone types, and a GUI scenario editor. Experiments cover three areas: (1) centralised vs independent value learning under varying partial observability, (2) sparse-reward exploration with RND, and (3) zero-shot generalisation via UED. The UED experiments find that agents generalise better to unseen zone layouts than to unseen unit specifications. Throughput benchmarks show advantages over SMAX when scenarios are frequently reconfigured.

**Compliance With Llm Reviewing Policy:**

Affirmed.

**Final Justification:**

The authors have addressed my initial concerns and I believe this work will have utility for the MARL research community.

**Key Questions For Authors:**

1. The SMAX throughput comparison in Figure 10 includes JIT recompilation overhead at every scenario reset. While this is a fair comparison in the UED setting, this overhead is not applicable in standard MARL training regimes. Would it be possible to provide a comparison for a fixed environment configuration so that it is clear how the environments compare when SMAX does not have JAX recompilations on each reset.
2. From my understanding, the primary mechanism enabling reconfiguration without recompilation appears to be padding arrays to a maximum size and masking inactive units. What prevents this pattern from being applied as a wrapper around SMAX?
3. The role-appropriate heuristic policy with multiple difficulty tiers is an interesting feature but, UED experiments fix opponents to novice-level. Using heuristic difficulty as a parameter for the UED study could yield interesting insights. Is this something the authors could provide?
4. I find the unit-spec vs. zone generalisation asymmetry interesting, but "unit specifications" combine qualitatively different parameters. Varying health likely poses a different generalisation challenge than varying speed or attack damage. Have the authors performed experimentation isolating individual unit parameters? This ablation would help identify where the difficulty in generalisation comes from.

**Limitations:**

No.

The authors provide a brief future work section but do not explicitly discuss limitations of the current work.

**Strengths And Weaknesses:**

## Strengths
* TABX contains many features along with a GUI, which form an easy-to-configure setting compared to most existing MARL environments. This reduces the barrier for designing custom MARL experiments and enables more experimental research that is not suitable in other settings.
* The ability to reset environment parameters without JIT recompilation is advantageous for UED and curriculum learning, where scenarios are often reconfigured during training.
* The experiments are organised around different research areas covering centralised value learning, exploration, and generalisation. The finding that agents generalise to unseen zone layouts but struggle with unseen unit specifications is an interesting empirical observation that could be utilised in future work.
* The paper clearly reports all hyperparameters, scenario descriptions and unit statistics, and the anonymised repository contains full implementations of all baselines.

## Weaknesses
 * The paper does not convincingly argue why TABX warrants adoption over extending SMAX in the non-UED setting. The primary throughput advantage stems from padding arrays to a fixed maximum size and masking inactive units to avoid recompilation.
* The SMAX throughput comparison in Figure 10 includes JIT recompilation overhead across 100 scenario resets. While this is reasonable for the UED setting, it is not clear what the speed gap is when compared in settings with static environment configs.
* The role-appropriate heuristic policy is presented as a key contribution; however, the UED experiments use fixed novel-level enemies and do not vary heuristic difficulty as a free parameter.
* While the environment is well-constructed, the core mechanics (battle simulator with heterogeneous units, terrain, partial observability) are already present in existing benchmarks like SMAX.

---

> ### Author Rebuttal · Authors · 2026-03-31
>
> We thank the reviewer for their supportive comments and helpful suggestions.
>
> **Q1./W2. The throughput comparison excluding JIT recompilation.**
>
> We provide the requested fixed-configuration throughput comparison. (https://imgur.com/a/D2hRqQs) TABX outperforms SMAX for smaller team sizes. For larger teams, SMAX is faster at lower parallelization scales due to the richer per-step dynamics of TABX, but this gap diminishes with increasing environment parallelism. This result confirms that the speedups observed in Figure 10 are primarily attributable to the elimination of JIT recompilation overhead rather than differences in raw throughput. We will include this figure in the revised manuscript.
>
> **Q2./W1. Reconfiguration without recompilation.**
>
> TABX indeed uses padding and masking to handle variable numbers of units and zones. However, this cannot be trivially replicated in SMAX via a simple wrapper. SMAX assumes a fixed environment configuration at initialization — unit types, counts, and their corresponding observation/action space dimensions are all determined at this stage and are tightly coupled throughout its internal state representation, step logic, and observation construction. Retrofitting SMAX to support dynamic reconfiguration would therefore require non-trivial refactoring of its core components rather than a surface-level modification. In contrast, TABX was designed from the ground up with reconfigurability as a first-class design principle, which enables seamless integration with UED pipelines and parallel training and evaluation across heterogeneous scenarios without recompilation.
>
> **Q3./W3. UED experiments across heuristic policy difficulty levels.**
>
> We evaluated the zero-shot generalization performance of various UED algorithms across a spectrum of heuristic policy tiers, ranging from random (highest noise) to expert (lowest noise). Episode win rates were averaged across four evaluation scenarios and three seeds. Heuristic policy configuration parameters include the stochasticity of actions and an aggressiveness threshold related to “kiting” behavior.
>
> As shown in https://imgur.com/a/k93Xmun and the table below, overall average performance is higher than in other free-parameter settings. This highlights that changes in unit attributes or environmental configurations are more critical for achieving robustness to unseen tasks. We will incorporate these extended training results in the revised manuscript.
>
> |Algorithm|Random|Novice|Medium|Advanced|Expert|Average|
> |-|-|-|-|-|-|-|
> |DR|1.00±0.00|0.95±0.04|0.87±0.14|0.83±0.17|0.84±0.16|0.90|
> |PLR|1.00±0.00|0.94±0.07|0.85±0.16|0.84±0.16|0.85±0.16|0.90|
> |PLR$^\perp$|1.00±0.00|0.84±0.20|0.70±0.28|0.64±0.29|0.66±0.28|0.77|
> |ACCEL|1.00±0.00|0.50±0.42|0.39±0.44|0.39±0.45|0.37±0.44|0.53|
> |SFL|1.00±0.00|0.94±0.06|0.86±0.15|0.83±0.18|0.81±0.23|0.89|
>
> **Q4. UED experiments isolating individual unit parameters.**
>
> Through isolated UED experiments, we identify Health as the primary bottleneck for generalization (https://imgur.com/a/5qPssBv). Unlike speed or attack metrics, health scaling introduces high variance in effective difficulty. This volatility disrupts the UED's curriculum generation, as the regret becomes too broad to navigate reliably. UED algorithms target the specific settings where an agent has the most to learn, which naturally forces agents to master a wide variety of different scenarios; however, high-dimensional unit specs create a search space so vast that the adaptive curriculum can no longer effectively guide the agent, resulting in training instability. With its granular control over unit and zone variables, TABX is an effective testbed for analyzing how MARL UED algorithms struggle with high-dimensional and wide-ranging parameter sets.
>
> **W4. In-depth comparison with existing benchmarks.**
>
> We would like to respectfully clarify that SMAX does not include terrain mechanics. Table 1 provides a detailed feature comparison between TABX and existing benchmarks. TABX differs in several key aspects:
>
> (1) TABX incorporates varied terrain and supporter units, necessitating more complex coordination strategies.
>
> (2) SMAX's heuristic policy behaves homogeneously across unit types, whereas our role-appropriate heuristic policy applies independent behavioral biases based on specific unit attributes across multiple difficulty tiers.
>
> (3) All of these components (unit compositions, terrain layouts, and opponent behaviors) are jointly reconfigurable at runtime without recompilation, which is a system-level design distinction rather than an incremental feature addition.
>
> (4) To show that TABX induces qualitatively distinct learning dynamics, we conducted direct comparisons of MAPPO and IPPO between TABX and SMAX on matched scenarios with aligned unit specifications and action dynamics. The results confirm that the differences extend beyond implementation to substantive behavioral complexity (see https://imgur.com/a/CjGIeFv for learning curves).

---

> > ### Author Rebuttal · Reviewer_uRhd · 2026-04-01
> >
> > Thank you for clearly answering my questions. Despite my initial misgivings, I do believe that this environment would be of high utility for researchers in the UED setting. The addition of the the additional throughput comparison will prevent confusion for standard RL.
> >
> > Due to this I have decided to change my score to a 5.

---

### Official Review · Reviewer_KPqY · 2026-03-13

**Soundness:** 3
**Presentation:** 3
**Significance:** 4
**Originality:** 3
**Overall Recommendation:** 5
**Confidence:** 3

**Summary:**

The paper introduces TABX, a JAX-based multi-agent simulator designed for reinforcement learning research. TABX provides flexibility in defining agents and environments, enabling researchers to conduct a wide range of experiments to evaluate their proposed MARL algorithms. The platform supports studies on centralized training, Unsupervised Environment Design (UED), zero-shot generalization, etc. It also removes the need to maintain multiple environment representations across experimental setups.

A notable contribution is the integration of GUI-assisted customization, which allows users to configure environments without deep familiarity with the codebase. The use of JAX enables efficient hardware parallelism and stable performance. The numerical experiments highlight and motivate the design features of the environment.

**Compliance With Llm Reviewing Policy:**

Affirmed.

**Final Justification:**

The paper presents a timely introduction to a simulation platform for addressing a diverse range of research questions in MARL.

**Key Questions For Authors:**

1. TABX involves a team of agents engaging in combat against opposing forces. From the description, the opposing team follows a fixed heuristic policy while the learning agents optimize against it. After training, is there a provision to pit two trained policies against each other (e.g., allies follow policy A and opponents follow policy B)? Allowing learned-policy-vs-learned-policy evaluation would significantly strengthen the framework, as it would enable direct comparison between MARL algorithms and make performance claims more compelling.

2. In Figure (4), the training curves appear not to be fully converged. Is it still appropriate to draw conclusions about algorithmic characteristics if convergence has not been reached? Would longer training runs change the relative comparisons?

3. References [1]–[2] appear to be missing from the related work discussion. In particular, [2] includes a battlefield-style environment designed for large-scale scenarios. A direct comparison or discussion would help better position TABX within the existing literature.

4. The conclusion of Figure (5) is not quite clear as the numbers are around the same magnitude – how can we deduce the relevance of global information here?

Minor Questions and Suggestions:

1. Is there a PyTorch implementation available, or are there plans to provide one?

2. Figure (1) is introduced in Lines 74–76, but its components are not explicitly described in the main text until later. Labeling parts (a)–(d) and briefly explaining each component would improve clarity.

3. In the main text, it would be helpful to summarize the scenarios used for the experiments (e.g., differences in environmental layouts and corresponding expected optimal behavior) and briefly state the reward structure. This would make it easier to interpret the results shown in Figure (4).

4. Is policy sharing required among agents? For example, in a two-farmer scenario, can the two agents use different policies (e.g., MADDPG without parameter sharing)? Explicit clarification of whether heterogeneous policies are supported would improve understanding.

References

[1] Zheng, Lianmin, et al. "Magent: A many-agent reinforcement learning platform for artificial collective intelligence." Proceedings of the AAAI conference on artificial intelligence. Vol. 32. No. 1. 2018.

[2] Terry, Jordan, et al. "Pettingzoo: Gym for multi-agent reinforcement learning." Advances in Neural Information Processing Systems 34 (2021): 15032-15043.

**Limitations:**

1. The environments appear primarily focused on cooperative MARL settings and does not discuss mixed cooperative–competitive scenarios.

2. As the number of agents increases, the framework may still be subject to scalability challenges associated with high-dimensional state and action spaces.

**Strengths And Weaknesses:**

Strengths

1. The paper presents a timely introduction of a multi-faceted simulation platform that can be utilized to answer a diverse range of research questions pertaining to MARL.

2. The reviewer particularly appreciates the introduction of a GUI-based customization framework, as it enhances accessibility for researchers who may not wish to modify core environment code.

3. The JAX-based parallelization enables efficient training, as reflected in the relatively short wall-clock times reported in the experiments.

4. The experimental section effectively demonstrates the motivation behind several design choices and features of the platform.

Weaknesses

1. From the current description, opponent agents appear to follow fixed heuristic policies. There does not seem to be a provision for directly pitting two learned RL policies against each other (e.g., MAPPO vs. IPPO). As a result, evaluation is limited to comparisons against a fixed heuristic baseline rather than policy-vs-policy comparisons.

2. Some important explanations (e.g., figure details, underlying physics of the environment) are absent/deferred to the appendix. While space constraints are understandable, moving key clarifications into the main text would improve readability and accessibility [see the suggestions written in the next section for exact details].

---

> ### Author Rebuttal · Authors · 2026-03-31
>
> We thank the reviewer for their thorough and constructive feedback.
>
> **Q1./W1. Policy v.s. policy comparison.**
>
> TABX directly supports this. Since the environment accepts actions for all agents across both teams and returns per-team rewards, two independently trained policies can be assigned to opposing teams without any modification to the core environment. This enables direct algorithm comparisons such as MAPPO vs IPPO in a competitive setting. As a demonstration, we provide a payoff matrix from such evaluations in https://imgur.com/a/OGFtMlT, and will include a discussion of this capability in the main text.
>
> **Q2. Train baselines with extended steps.**
>
> For scenarios where convergence was insufficient, we extended training to 50M steps (crossfire, vsrangers, ambush, superking) and 100M steps (clover, bypass, ribbon, grid) in https://imgur.com/a/iJ62Tgd. The results from these extended runs are consistent with the algorithmic comparisons reported in the main text, supporting the validity of our conclusions. We will include these updated training details in the revised manuscript.
>
> | Algorithm   | Crossfire         | Vsrangers         | Ambush            | Superking         | Clover            | Bypass            | Ribbon            | Grid              |
> |:------------|:------------------|:------------------|:------------------|:------------------|:------------------|:------------------|:------------------|:------------------|
> | MAPPO       | 0.961 ± 0.020     | 0.953 ± 0.032     | 0.797 ± 0.101     | **0.583 ± 0.016** | **0.997 ± 0.003** | **0.659 ± 0.131** | **0.966 ± 0.010** | **0.331 ± 0.026** |
> | IPPO        | **0.997 ± 0.003** | 0.995 ± 0.003     | **0.995 ± 0.005** | 0.565 ± 0.061     | 0.724 ± 0.249     | 0.422 ± 0.112     | 0.367 ± 0.198     | 0.023 ± 0.005     |
> | QMIX        | 0.732 ± 0.237     | **1.000 ± 0.000** | 0.878 ± 0.107     | 0.107 ± 0.099     | 0.995 ± 0.003     | 0.120 ± 0.073     | 0.299 ± 0.292     | 0.141 ± 0.068     |
> | IQL         | 0.461 ± 0.102     | 0.987 ± 0.005     | 0.779 ± 0.099     | 0.385 ± 0.049     | 0.227 ± 0.106     | 0.091 ± 0.046     | 0.018 ± 0.011     | 0.008 ± 0.000     |
>
> **Q3. Missing related work.**
>
> Thank you for identifying the missing related works. We will incorporate these benchmarks into the Related Work section. PettingZoo [2] provides a diverse set of multi-agent environments following the Agent-Environment Cycle (AEC) model. Although this benchmark handles a large number of agents, it still suffers from static scenario sets and the overhead of slow computation times. On the other hand, MAgent [1] supports a highly scalable framework on GPU servers and enables flexible environment design. However, unlike TABX, the agents in MAgent are homogeneous and unconfigurable regarding unit specifications.
>
> **Q4. Deduction the relevance of global information in Figure 5.**
>
> We thank the reviewer for raising this point. While the absolute difference may appear small, we note that the undiscounted return ranges over $[-2.0, 2.0]$. Since the win/loss reward ($\pm 1$) is given only at the final timestep, its contribution to the discounted return is scaled by $\gamma^{400} \approx 0.018$, where 400 is an average of episode lengths. making the effective range of discounted return substantially narrower. Within this compressed range, the observed gap of ${\sim}0.2$ is statistically significant, supporting the conclusion that value error leads to a meaningful performance advantage.
>
> To further substantiate this, we computed the Pearson correlation between the episode return difference and value error difference (IPPO - MAPPO) across all scenarios. When aggregating over seeds, the correlation is strongly negative (r = -0.819, p = 0.0128), confirming that lower value error is reliably associated with higher returns at the scenario level. We additionally report the correlation using individual seed-level data points without averaging (r = -0.681, p < 0.0001), which corroborates the same trend. Scatter plots for both analyses are provided in https://imgur.com/a/5rpM9J0.
>
> **Minor Q4. Homogeneous policy.**
>
> Policy sharing is not required in TABX. The environment API accepts actions from each agent independently, and each agent's observation includes its own unit-specific information. This naturally supports heterogeneous policies, allowing agents with different unit specifications (e.g., combat units vs. supporters) to be trained with distinct policies.
>
> **Minor Qs./W2.**
>
> Thank you for suggesting areas of improvement for our paper. We will revise our manuscript based on your advice. While we recommend using TABX with JAX implementations to maintain high throughput, researchers can also use TABX with frameworks such as PyTorch.

---

> > ### Author Rebuttal · Reviewer_KPqY · 2026-04-03
> >
> > The authors have addressed my main concerns.

---

### Official Review · Reviewer_nK4L · 2026-03-13

**Soundness:** 3
**Presentation:** 3
**Significance:** 3
**Originality:** 3
**Overall Recommendation:** 5
**Confidence:** 4

**Summary:**

This paper introduces TABX (Totally Accelerated Battle Simulator in JAX), a GPU accelerated, configurable multiagent environment for cooperative MARL. It provides a combination of JAX based gpu acceleration with unit specifications, terrain zones, heuristic opponent policies, and physics dynamics (modifiable without code changes or JIT recompilation). It features partial observability, hurtbox attack mechanics, heterogeneous unit roles, and terrain zones (lava, bush, swamp). The authors provide a GUI editor and a suite of predefined scenarios. Experiments demonstrate three use cases: (1) centralized vs independent value learning under varying partial observability, (2) exploration in sparse-reward long-horizon tasks, and (3) zero-shot generalization via unsupervised environment design (UED). Benchmarks show near log-linear throughput scaling and substantial speedups over SMAX when frequent scenario changes are needed.

**Compliance With Llm Reviewing Policy:**

Affirmed.

**Final Justification:**

The authors have clarified some of the limitations, I expect the authors to add this reflection in the final version.

**Key Questions For Authors:**

- Can you provide a direct algorithmic comparison on matched scenarios between TABX and SMAX? For example, running the same MARL algorithms on comparable team compositions in both environments could help show that TABX additional mechanics (hurtbox, terrain, fan-shaped FoV) lead to qualitatively different learning dynamics or just added complexity. I basically want to assess whether TABX leads to novel insights/results compared to SMAX. This is besides the advantage of fast jit-compilation between env configurations (which we might be able to implement in SMAX too?).
- The UED experiments reveal that generalization over unit specifications is harder than over zones, can you provide a deeper analysis of why? The current explanation (unit specs affect controllable dynamics) is intuitive but surface-level. Eg do certain unit attribute dimensions (health vs speed vs damage) contribute differently to the generalization gap?
- Have you considered self-play or learnt opponent policies (is the env compatible with this?)? The current heuristic opponent setup limits the environment to only cooperative evaluation. I understand that this is likely intentional, given the relatedness to other benchmarks like SMAX, and for reproducibility, but the possibility for self-play etc. would seem really useful for the community.

**Limitations:**

The authors include a brief impact statement but do not discuss limitations of their work in a dedicated section. (Eg, what is TABX not suited for?).

**Strengths And Weaknesses:**

Strenghts

TABX addresses a gap in MARL benchmarks by combining hardware acceleration, rich configurations, and multi-agent support in a single framework. Table 1 makes the case that no existing benchmark offers all of these properties simultaneously. The ability to reconfigure environment parameters without JIT recompilation is practical, also for curriculum learning research, and the speed comparison against SMAX shows this advantage.
The scenario design is well-motivated. The authors construct scenarios that isolate specific MARL settings, eg crossfire/vsrangers where local observations suffice versus clover/ribbon where global state information is required for value estimation. The value estimation error provides an improved evaluation compared to win rate alone and supports the claims about centralized vs. independent learning. The exploration scenarios (pingpong, encirclement) with their specific reward design are also well-motivated.
The GUI scenario editor makes it more accessible to design custom scenarios. The paper is thorough in the appendices, with extensive detail on observation spaces, interaction dynamics, heuristic policy mechanisms, and hyperparameters, supporting reproducibility.

Weaknesses:
The experiments, while adequate for demonstrating the environment's capabilities, remain illustrative rather than generating novel insights about MARL algorithms. The findings that IPPO outperforms MAPPO with local observations, and that MAPPO benefits when using global information, are well-established in prior work (the authors themselves cite De Witt et al., 2020). Similarly, the observation that RND helps in sparse reward settings is not surprising. The UED experiments show generalization for unit specifications is more difficult than for zone layouts. This is an interesting observation but is not really deeply analyzed. I believe the paper would benefit from at least one experiment that shows a genuinely new or counterintuitive finding through TABX’s specific properties.
The combat rewards (health ratio differences, win/loss) and the mechanics can limit the types of cooperation behaviors that can be studied. Coordination challenges in TABX are mostly about "who should I attack”, “where to position”, “when to heal". This can lead to insights, but is not representative of the full range of cooperative MARL problems (e.g. resource sharing, task allocation, communication emergence). This is not a major shortcoming on its own, but I believe the limitations/boundaries of this benchmark could be acknowledged more explicitly in the paper.
Moreover, I believe the comparison with existing benchmarks could be stronger. The paper compares throughput against SMAX but does not provide any algorithmic performance comparisons for analogous scenarios (e.g. comparing win rates or learning curves on similar team settings in SMAX vs TABX). This makes it difficult to assess whether the added complexity of TABX mechanics (hurtboxes, fan-shaped FoV, terrain) leads to qualitatively different or more challenging learning problems versus simply being more complex to implement.
The novelty of the paper lies mainly in systems/engineering design. While this is valuable, the individual components (JAX acceleration, configurable parameters, hurtbox mechanics, terrain zones) are not themselves new. The scientific contributions beyond the engineering  contributions seem rather limited.

Soundness

The technical implementation appears correct, and the benchmark could be useful to the community. The methodology seems correct (5 seeds, standard errors reported), and the claims are well-supported in general. The value estimation error analysis is a nice addition. However, the experiments mainly confirm known results rather than truly testing new hypotheses.

Presentation
The paper is well-written and clearly structured. Figures are informative, and the appendices are thorough. The scenario are clearly described and well-motivated. Minor issues: the main paper is dense with environment description, leaving limited space for experimental analysis.

Significance
While the engineering contribution is solid and I see the environment potentially getting adoption, the paper is limited by it seemingly being an engineering upgrade to SMAX. The experiments mainly serve as reiterating known results within a new environment rather than advancing understanding of MARL .

Originality
The individual components are taken from well-known practices (JAX acceleration from JaxMARL, battle mechanics from StarCraft environments, terrain from various game engines, UED). The combination is well-executed but the paper does not strongly introduce new insights into marl research. The hurtbox-based non-targeted interaction is a nice design choice.

---

> ### Author Rebuttal · Authors · 2026-03-31
>
> We are grateful for the reviewer's thorough and constructive feedback.
>
> **Q1./W3. Comparing win rates or learning curves on similar team settings in SMAX vs TABX.**
>
> We thank the reviewer for this suggestion and conducted direct comparisons of MAPPO and IPPO between TABX and SMAX on matched scenarios, where we aligned unit specifications and action dynamics to match SMAX as closely as possible. The enemy heuristic policy was set to medium, as in our main experiments. Learning curves are in https://imgur.com/a/CjGIeFv.
>
> TABX proves harder to solve than SMAX in most scenarios, which we attribute to its more complex low-level dynamics — agents must learn precise positioning and orientation rather than simply closing the distance as in SMAX. In _2s3z_, melee units face additional difficulty as the non-targeting attack mechanism requires them to physically close in and land hits to receive any learning signal, and in _6h_vs_8z_, agents must simultaneously manage distance, rotation, and attack timing. The exception is _5m_vs_6m_, where TABX initially underperforms SMAX due to the difficulty of learning these dynamics, but surpasses SMAX after approximately 2M steps. These results demonstrate that TABX induces qualitatively distinct learning dynamics from SMAX, rather than simply adding implementation complexity.
>
> **Q2. Generalization challenges regarding diverse unit specifications.**
>
> We identify Health as the dominant bottleneck for generalization through isolated UED experiments on individual unit attributes (https://imgur.com/a/5qPssBv). Health induces a substantially wider effective difficulty range than speed or attack damage, creating a broad regret landscape that interacts poorly with UED's adaptive curriculum. UED methods prioritize environment parameters with the highest estimated learning progress, naturally expanding coverage over diverse configurations; however, when the underlying parameter space is both high-dimensional and broad in range — as is the case with unit specifications — this adaptive mechanism becomes misaligned with what the agent can effectively learn, destabilizing training. The contrast between zone-level and unit-level results further corroborates this finding: unit specifications scale more aggressively than zone variations due to the larger number of controllable entities, producing a higher-dimensional configuration space where existing UED methods struggle. More broadly, these results suggest that our environment, with its independently tunable zone- and unit-level parameters, can serve as a useful testbed for diagnosing and studying the limitations of MARL UED algorithms under varying dimensionality and range conditions.
>
> **Q3. Self-play or learned opponent policy.**
>
> TABX is fully compatible with both self-play and learned opponent policies. The environment accepts actions for all agents across both teams and returns per-team rewards, making self-play and policy-vs-policy evaluation straightforward. The heuristic opponent is implemented as a separate wrapper, which can be replaced with any learned policy without modifying the core environment. As a demonstration, we provide a payoff matrix obtained from training with learned opponent policies in https://imgur.com/a/OGFtMlT. We will include a discussion of this capability in the main text.
>
> **W1. Explicit acknowledgement of TABX's limitations/boundaries and clarification of scientific contributions beyond engineering design.**
>
> Although TABX offers a flexible framework for multi-agent research, we acknowledge its current scope as a limitation. As one promising direction for generating novel insights, we are exploring a transfer learning paradigm in which a single agent first masters low-level control before being deployed into a multi-agent setting. This approach could allow agents to decouple the challenges of individual control, which are more demanding in TABX due to the non-targeted attack mechanism, from those of multi-agent cooperation. More broadly, hierarchical decomposition of cooperative tasks represents a fruitful avenue for future work with TABX, and we plan to investigate this direction further.
>
> **W2. Addressing the limited range of cooperative MARL problems.**
>
> We acknowledge this limitation. TABX is intentionally designed as a combat-oriented benchmark, and as such, the range of cooperative behaviors it supports is inherently scoped to that domain. We will add a dedicated Limitations section to the paper that explicitly discusses this boundary, including the types of cooperative MARL problems (e.g., resource sharing, task allocation, communication emergence) that TABX is not designed to address.

---

> > ### Author Rebuttal · Reviewer_nK4L · 2026-04-03
> >
> > Thank you for answering my questions. The simulator is well-engineerd, and I expect it will get adopted by the community. I therefore recommend acceptance."

---

### Decision · Program_Chairs · 2026-04-30

**Decision:**

Accept (regular)

**Comment:**

This paper introduces the TABX battle simulator in JAX, targeted at filling a gap in the existing simulator suite (e.g., SMAC) for simulators that are both fast (hardware accelerated, parallelizable) and easily and efficiently configurable w/out JIT recompilation (GUI, heterogeneous unit policies, useful for UED). The reviewers agreed that the “environment would be of high utility for researchers”, so we are recommending acceptance. Lastly, I would encourage the authors to re-consider their impact statement in light of the title, environment domains, and “battlefield” language. Recall the link to the [blog post](https://medium.com/@GovAI/a-guide-to-writing-the-neurips-impact-statement-4293b723f832) on ICML’s CfP, in particular, how a military audience might interpret this work.